# FSP-LAPLACE: Function-Space Priors for the Laplace Approximation in Bayesian Deep Learning

**Tristan Cinquin**[*]
Tübingen AI Center, University of Tübingen
`tristan.cinquin@uni-tuebingen.de`

**Marvin Pförtner**[*]
Tübingen AI Center, University of Tübingen
`marvin.pfoertner@uni-tuebingen.de`

**Vincent Fortuin**
Helmholtz AI, TU Munich
`vincent.fortuin@tum.de`

**Philipp Hennig**
Tübingen AI Center, University of Tübingen
`philipp.hennig@uni-tuebingen.de`

**Robert Bamler**
Tübingen AI Center, University of Tübingen
`robert.bamler@uni-tuebingen.de`

## Abstract

Laplace approximations are popular techniques for endowing deep networks with epistemic uncertainty estimates as they can be applied without altering the predictions of the trained network, and they scale to large models and datasets. While the choice of prior strongly affects the resulting posterior distribution, computational tractability and lack of interpretability of the weight space typically limit the Laplace approximation to isotropic Gaussian priors, which are known to cause pathological behavior as depth increases. As a remedy, we directly place a prior on function space. More precisely, since Lebesgue densities do not exist on infinite-dimensional function spaces, we recast training as finding the so-called weak mode of the posterior measure under a Gaussian process (GP) prior restricted to the space of functions representable by the neural network. Through the GP prior, one can express structured and interpretable inductive biases, such as regularity or periodicity, directly in function space, while still exploiting the implicit inductive biases that allow deep networks to generalize. After model linearization, the training objective induces a negative log-posterior density to which we apply a Laplace approximation, leveraging highly scalable methods from matrix-free linear algebra. Our method provides improved results where prior knowledge is abundant (as is the case in many scientific inference tasks). At the same time, it stays competitive for black-box supervised learning problems, where neural networks typically excel.

## 1 Introduction

Neural networks (NNs) have become the workhorse for many machine learning tasks, but they do not quantify the uncertainty arising from data scarcity—the *epistemic* uncertainty. NNs therefore cannot estimate their confidence in their predictions, which is needed for safety-critical applications, decision making, and scientific modeling [1–3]. As a solution, Bayesian neural networks (BNNs) cast training as approximating the Bayesian posterior distribution over the weights (i.e., the distribution of the model parameters given the training data), thus naturally capturing epistemic uncertainty. Various methods exist for approximating the (intractable) posterior, either by a set of samples (e.g., MCMC) or

---

[*]Equal contribution

38th Conference on Neural Information Processing Systems (NeurIPS 2024).

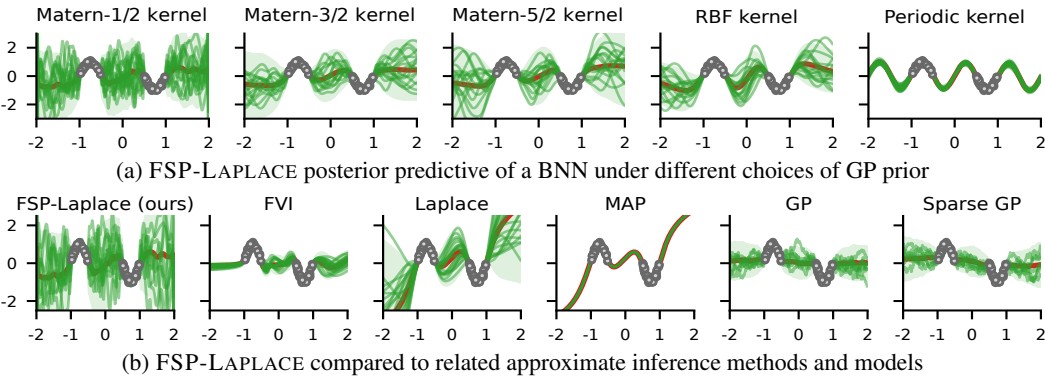

(a) FSP-LAPLACE posterior predictive of a BNN under different choices of GP prior

(b) FSP-LAPLACE compared to related approximate inference methods and models

Figure 1: FSP-LAPLACE allows for efficient approximate Bayesian neural network (BNN) inference under interpretable function space priors. Using our method, it is possible to encode functional properties like smoothness, lengthscale, or periodicity through a Gaussian process (GP) prior. The gray data points in the plots are noisy observations of a periodic function.

by a parametric distribution (e.g., variational inference or Laplace approximations). While sampling methods can be asymptotically exact for large numbers of samples, they are typically expensive for large models, whereas variational inference (VI) and the Laplace approximation are more scalable. The Laplace approximation is particularly appealing as it does not alter maximum-a-posteriori (MAP) predictions. Compared to VI, its differential nature allows the Laplace approximation to scale to large datasets and models, and it has been shown to provide well-calibrated uncertainty estimates [4, 5].

**The need for function-space priors in BNNs.** While the choice of prior strongly influences the uncertainty estimates obtained from the posterior [6, 7], there are hardly any methods for elicitation of informative priors in the literature on BNNs, with the notable exceptions of Sun et al. [8] and Tran et al. [9]. This is not just a conceptual issue. For instance, the default isotropic Gaussian prior, commonly thought of as uninformative, actually carries incorrect beliefs about the posterior (uni-modality, independent weights, etc.) [6, 10] and has known pathologies in deep NNs [9]. As network weights are not interpretable, formulating a good prior on them is virtually impossible. Current methods work around this issue by model selection, either through expensive cross-validation, or type-II maximum likelihood estimation with a Laplace approximation under an isotropic Gaussian prior, which can only be computed exactly for small networks, and has known issues with normalization layers [11, 12]. As a solution, Sun et al. [8] proposed a VI method to specify priors directly on the function implemented by the BNN, with promising results. Function-space priors incorporate interpretable knowledge about the variance, regularity, periodicity, or length scale, building on the extensive Gaussian Process (GP) literature [13]. But it turns out that the method by Sun et al. [8] is difficult to use in practice: the Kullback-Leibler (KL) divergence regularizing VI is infinite for most function-space priors of interest [14], and estimating finite variants of it is challenging [15].

This paper addresses the lack of a procedure to specify informative priors in the Laplace approximation by proposing a method to use interpretable GP priors. Motivated by MAP estimation theory, we first derive an objective function that regularizes the neural network in function space using a GP prior, and whose minimizer corresponds to the MAP estimator of a GP on the space of functions represented by the network. We then apply the Laplace approximation to the log-posterior density induced by the objective, allowing us to effectively incorporate beliefs from a GP prior into a deep network (see Figure 1). Efficient matrix-free methods from numerical linear algebra allow scaling our method to large models and datasets. We show the effectiveness of our method by showing improved results in cases where prior knowledge is available, and competitive performance for black-box regression and classification tasks, where neural networks typically excel. We make the following contributions:

1. We propose a novel objective function for deep neural networks that allows incorporating interpretable prior knowledge through a GP prior in function space.

2. We develop an efficient and scalable Laplace approximation that endows the neural network with epistemic uncertainty reflecting the beliefs specified by the GP prior in function space.

3. A range of experiments shows that our method improves performance on tasks with prior knowledge in the form of a kernel, while showing competitive performance for black-box regression, classification, out-of-distribution detection, and Bayesian optimization tasks.

## 2  Preliminaries: Laplace approximation in weight space

For a given dataset $\mathbb{D} = (\boldsymbol{X}, \boldsymbol{Y})$ of inputs $\boldsymbol{X} = (\boldsymbol{x}^{(i)})_{i=1}^n \in \mathbb{X}^n$ and targets $\boldsymbol{Y} = (\boldsymbol{y}^{(i)})_{i=1}^n \in \mathbb{Y}^n$, we consider a model of the data that is parameterized by a neural network $\boldsymbol{f} \colon \mathbb{X} \times \mathbb{W} \to \mathbb{R}^{d'}$ with weights $\boldsymbol{w} \in \mathbb{W} \subset \mathbb{R}^p$, a likelihood $p(\boldsymbol{Y} \mid \boldsymbol{f}(\boldsymbol{X}, \boldsymbol{w})) \coloneqq \prod_{i=1}^n p(\boldsymbol{y}^{(i)} \mid \boldsymbol{f}(\boldsymbol{x}^{(i)}, \boldsymbol{w}))$, and a prior $p(\boldsymbol{w})$.

We seek the Bayesian posterior $p(\boldsymbol{w} \mid \mathbb{D}) \propto p(\boldsymbol{Y} \mid \boldsymbol{f}(\boldsymbol{X}, \boldsymbol{w}))\, p(\boldsymbol{w})$. As it is intractable in BNNs, approximate inference methods have been developed. Among these, the Laplace approximation [16] to the posterior is given by a Gaussian distribution $q(\mathbf{w} = \boldsymbol{w}) = \mathcal{N}\left(\boldsymbol{w}; \boldsymbol{w}^\star, \boldsymbol{\Lambda}^{-1}\right)$, whose parameters are found by MAP estimation $\boldsymbol{w}^\star \in \arg\min_{\boldsymbol{w} \in \mathbb{W}} R_{\mathrm{WS}}(\boldsymbol{w})$ of the weights, where

$$R_{\mathrm{WS}}(\boldsymbol{w}) \coloneqq -\log p(\boldsymbol{w} \mid \mathbb{D}) = -\log p(\boldsymbol{w}) - \sum_{i=1}^n \log p(\boldsymbol{y}^{(i)} \mid \boldsymbol{f}(\boldsymbol{x}^{(i)}, \boldsymbol{w})) + \text{const.} \qquad (2.1)$$

and computing the Hessian of the negative log-posterior $\boldsymbol{\Lambda} \coloneqq -\left.\mathbf{H}_{\boldsymbol{w}} \log p(\boldsymbol{w} \mid \mathbb{D})\right|_{\boldsymbol{w}=\boldsymbol{w}^\star} \in \mathbb{R}^{p \times p}$.

**The linearized Laplace approximation.** Computing $\boldsymbol{\Lambda}$ involves the Hessian of the neural network w.r.t. its weights which is generally expensive. To make the Laplace approximation scalable, it is common to linearize the network around $\boldsymbol{w}^\star$ before computing $\boldsymbol{\Lambda}$ [17],

$$\boldsymbol{f}(\boldsymbol{x}, \boldsymbol{w}) \approx \boldsymbol{f}(\boldsymbol{x}, \boldsymbol{w}^\star) + \boldsymbol{J}_{\boldsymbol{w}^\star}(\boldsymbol{x})(\boldsymbol{w} - \boldsymbol{w}^\star) =: \boldsymbol{f}^{\mathrm{lin}}(\boldsymbol{x}, \boldsymbol{w}) \qquad (2.2)$$

with Jacobian $\boldsymbol{J}_{\boldsymbol{w}^\star}(\boldsymbol{x}) = \left.\mathbf{D}_{\boldsymbol{w}}\, \boldsymbol{f}(\boldsymbol{x}, \boldsymbol{w})\right|_{\boldsymbol{w}=\boldsymbol{w}^\star}$. Thus, the approximate posterior precision matrix $\boldsymbol{\Lambda}$ is

$$\boldsymbol{\Lambda} \approx -\left.\mathbf{H}_{\boldsymbol{w}} \log p(\boldsymbol{w})\right|_{\boldsymbol{w}=\boldsymbol{w}^\star} - \sum_{i=1}^n \boldsymbol{J}_{\boldsymbol{w}^\star}(\boldsymbol{x}^{(i)})^\top \boldsymbol{H}_{\boldsymbol{w}^\star}^{(i)} \boldsymbol{J}_{\boldsymbol{w}^\star}(\boldsymbol{x}^{(i)}), \qquad (2.3)$$

where $\boldsymbol{H}_{\boldsymbol{w}^\star}^{(i)} = \left.\mathbf{H}_{\boldsymbol{f}}\, \log p(\boldsymbol{y}^{(i)} \mid \boldsymbol{f})\right|_{\boldsymbol{f}=\boldsymbol{f}(\boldsymbol{x}^{(i)}, \boldsymbol{w}^\star)}$. Thus, under the linearized network, the Hessian of the negative log-likelihood (NLL) coincides with the *generalized Gauss-Newton matrix* (GGN) of the NLL. Crucially, the GGN is positive-(semi)definite even if $-\left.\mathbf{H}_{\boldsymbol{w}} \log p(\boldsymbol{Y} \mid \boldsymbol{f}(\boldsymbol{X}, \boldsymbol{w}))\right|_{\boldsymbol{w}=\boldsymbol{w}^\star}$ is not.

## 3  FSP-LAPLACE: Laplace approximation under function-space priors

A conventional Laplace approximation in neural networks requires a prior in weight space, with the issues discussed in Section 1. We now present FSP-LAPLACE, a method for computing Laplace approximations under interpretable GP priors in function space. Section 3.1 introduces an objective function that is a log-density under local linearization. Section 3.2 proposes a scalable algorithm for the linearized Laplace approximation with a function-space prior using matrix-free linear algebra.

### 3.1  Laplace approximations in function space

We motivate our Laplace approximation in function space through the lens of MAP estimation in an (infinite-dimensional) function space under a GP prior. As Lebesgue densities do not exist in (infinite-dimensional) function spaces, we cannot minimize Equation (2.1) to find the MAP estimate. We address this issue using a generalized notion of MAP estimation, resulting in a minimizer of a regularized objective on the reproducing kernel Hilbert space $\mathbb{H}_{\boldsymbol{\Sigma}}$. We then constrain the objective to the set of functions representable by the neural network and minimize it using tools from deep learning. Finally, local linearization of the neural network turns this objective into a valid negative log-density, from which we obtain the posterior covariance by computing its Hessian.

**MAP estimation in neural networks under Gaussian process priors.** The first step of the Laplace approximation is to find a MAP estimate of the neural network weights $\boldsymbol{w}$, i.e., the minimizer of the negative log-density function in Equation (2.1) w.r.t. the Lebesgue measure as a "neutral" reference measure. In our method, we regularize the neural network in function space using a $d'$-output GP prior $\mathbf{f} \sim \mathcal{GP}\left(\boldsymbol{\mu}, \boldsymbol{\Sigma}\right)$ with index set $\mathbb{X}$. However, a (nonparametric) GP takes its values in an infinite-dimensional (Banach) space $\mathbb{B}$ of functions, where no such reference Lebesgue measure exists [18]. Rather, the GP induces a prior *measure* $\mathrm{P}_{\mathbb{B}}$ on $\mathbb{B}$ [19, Section B.2]. We are thus interested in the "mode" of the posterior measure $\mathrm{P}_{\mathbb{B}}^{\boldsymbol{Y}}$ under $\mathrm{P}_{\mathbb{B}}$ defined by the Radon-Nikodym

derivative $P_{\mathbb{B}}^{\boldsymbol{Y}}(\mathrm{d}\boldsymbol{f}) \propto \exp\left(-\Phi^{\boldsymbol{Y}}(\mathrm{d}\boldsymbol{f})\right) P_{\mathbb{B}}(\mathrm{d}\boldsymbol{f})$ where $\Phi^{\boldsymbol{Y}}$ is the *potential*, in essence the negative log-likelihood functional of the model. In our case, $\Phi^{\boldsymbol{Y}}(\boldsymbol{f}) = -\sum_{i=1}^{n} \log p(\boldsymbol{y}^{(i)} \mid \boldsymbol{f}(\boldsymbol{x}^{(i)}))$. Similar to Bayes' rule in finite dimensions, this Radon-Nikodym derivative relates the prior measure to the posterior measure by reweighting. Since there is no Lebesgue measure, the (standard) mode is undefined, and we therefore follow Lambley [20], using so-called *weak modes*.

**Definition 1** (Weak Mode [see e.g., 20, Definition 2.1]). Let $\mathbb{B}$ be a separable Banach space and let $P$ be a probability measure on $(\mathbb{B}, \mathcal{B}(\mathbb{B}))$. A *weak mode* of $P$ is any point $\boldsymbol{f}^{\star} \in \operatorname{supp} P$ such that

$$\limsup_{r\downarrow 0} \frac{P(B_r(\boldsymbol{f}))}{P(B_r(\boldsymbol{f}^{\star}))} \leq 1 \qquad \text{for all } \boldsymbol{f} \in \mathbb{B}. \tag{3.1}$$

Above, $\mathcal{B}(\mathbb{B})$ denotes the Borel $\sigma$-algebra on $\mathbb{B}$ and $B_r(\boldsymbol{f}) \subset \mathbb{B}$ is an open ball with radius $r$ centered at a point $\boldsymbol{f} \in \mathbb{B}$. The intuition for the weak mode is the same as for the finite-dimensional mode (indeed they coincide when a Lebesgue measure exists), but the weak mode generalizes this notion to infinite-dimensional (separable) Banach spaces. Under certain technical assumptions on the potential $\Phi^{\boldsymbol{Y}}$ (see Assumption A.2 in the appendix), Lambley [20] shows that any solution to

$$\arg\min_{\boldsymbol{f}\in\mathbb{H}_{\boldsymbol{\Sigma}}} \underbrace{\Phi^{\boldsymbol{Y}}(\boldsymbol{f}) + \frac{1}{2}\|\boldsymbol{f} - \boldsymbol{\mu}\|_{\mathbb{H}_{\boldsymbol{\Sigma}}}^2}_{=:R_{\mathrm{FSP}}(\boldsymbol{f})} \tag{3.2}$$

is a weak mode of the posterior probability measure $P_{\mathbb{B}}^{\boldsymbol{Y}}$.

Equation (3.2) casts the weak mode of the posterior measure $P_{\mathbb{B}}^{\boldsymbol{Y}}$ as the solution of an optimization problem in the RKHS $\mathbb{H}_{\boldsymbol{\Sigma}}$. We can now relate it to an optimization problem in weight space $\mathbb{W}$, and apply tools from deep learning. Informally speaking, we assume that the intersection of the set of functions represented by the neural network $\mathbb{F} := \{\boldsymbol{f}(\,\cdot\,, \boldsymbol{w}): \boldsymbol{w} \in \mathbb{W}\} \subset \mathbb{B} \subset (\mathbb{R}^{d'})^{\mathbb{X}}$ and $\mathbb{H}_{\boldsymbol{\Sigma}}$ is non-empty, and constrain the optimization problem in Equation (3.2) such that $\boldsymbol{f}$ belongs to both sets

$$\arg\min_{\boldsymbol{f}\in\mathbb{H}_{\boldsymbol{\Sigma}}\cap\mathbb{F}} R_{\mathrm{FSP}}(\boldsymbol{f}). \tag{3.3}$$

Unfortunately, the framework by Lambley [20] cannot give probabilistic meaning to optimization problems with hard constraints of the form $\boldsymbol{f} \in \mathbb{F}$. To address this, we adopt and elaborate on the informal strategy from Chen et al. [21, Remark 2.4]. Denote by $d_{\mathbb{B}}(\boldsymbol{f}, \mathbb{F}) := \inf_{\boldsymbol{f}_0 \in \mathbb{F}}\|\boldsymbol{f}_0 - \boldsymbol{f}\|_{\mathbb{B}}$ the distance of a function $\boldsymbol{f} \in \mathbb{B}$ to the set $\mathbb{F} \subset \mathbb{B}$. Under Assumption A.3, $d_{\mathbb{B}}(\boldsymbol{f}, \mathbb{F}) = 0$ if and only if $\boldsymbol{f} \in \mathbb{F}$ by Lemma A.2. Hence, we can relax the constraint $\boldsymbol{f} \in \mathbb{F}$ by adding $\frac{1}{2\lambda^2} d_{\mathbb{B}}^2(\boldsymbol{f}, \mathbb{F})$ with $\lambda > 0$ to the objective in Equation (3.2). Intuitively, the resulting optimization problem is the MAP problem for the measure $P_{\mathbb{B}}^{\boldsymbol{Y},\lambda}$ obtained by conditioning $P_{\mathbb{B}}^{\boldsymbol{Y}}$ on the observation that $d_{\mathbb{B}}(\mathbf{f}, \mathbb{F}) + \epsilon_{\lambda} = 0$, where $\epsilon_{\lambda}$ is independent centered Gaussian measurement noise with variance $\lambda^2$. Formally:

**Proposition 1.** *Let Assumptions A.1 to A.3 hold. For $\lambda > 0$, define $\Phi^{\boldsymbol{Y},\lambda}: \mathbb{B} \to \mathbb{R}, \boldsymbol{f} \mapsto \Phi^{\boldsymbol{Y}}(\boldsymbol{f}) + \frac{1}{2\lambda^2} d_{\mathbb{B}}^2(\boldsymbol{f}, \mathbb{F})$. Then the posterior measure $P_{\mathbb{B}}^{\boldsymbol{Y},\lambda}(\mathrm{d}\boldsymbol{f}) \propto \exp\left(-\Phi^{\boldsymbol{Y},\lambda}(\mathrm{d}\boldsymbol{f})\right) P_{\mathbb{B}}(\mathrm{d}\boldsymbol{f})$ has at least one weak mode $\boldsymbol{f}^{\star} \in \mathbb{H}_{\boldsymbol{\Sigma}}$, and the weak modes of $P_{\mathbb{B}}^{\boldsymbol{Y},\lambda}$ coincide with the minimizers of*

$$R_{FSP}^{\lambda}: \mathbb{H}_{\boldsymbol{\Sigma}} \to \mathbb{R}, \boldsymbol{f} \mapsto \Phi^{\boldsymbol{Y},\lambda}(\boldsymbol{f}) + \frac{1}{2}\|\boldsymbol{f} - \boldsymbol{\mu}\|_{\mathbb{H}_{\boldsymbol{\Sigma}}}^2. \tag{3.4}$$

As $\lambda \to 0$, the term $\frac{1}{2\lambda^2} d_{\mathbb{B}}^2(\boldsymbol{f}, \mathbb{F})$ forces the minimizers of $R_{\mathrm{FSP}}^{\lambda}$ to converge to functions in $\mathbb{H}_{\boldsymbol{\Sigma}} \cap \mathbb{F}$ that minimize $R_{\mathrm{FSP}}$:

**Proposition 2.** *Let Assumptions A.1 to A.3 hold. Let $\{\lambda_n\}_{n\in\mathbb{N}} \subset \mathbb{R}_{>0}$ with $\lambda_n \to 0$, and $\{\boldsymbol{f}_n^{\star}\}_{n\in\mathbb{N}} \subset \mathbb{H}_{\boldsymbol{\Sigma}}$ such that $\boldsymbol{f}_n^{\star}$ is a minimizer of $R_{FSP}^{\lambda_n}$. Then $\{\boldsymbol{f}_n^{\star}\}_{n\in\mathbb{N}}$ has an $\mathbb{H}_{\boldsymbol{\Sigma}}$-weakly convergent subsequence with limit $\boldsymbol{f}^{\star} \in \mathbb{H}_{\boldsymbol{\Sigma}} \cap \mathbb{F}$. Moreover, $\boldsymbol{f}^{\star}$ is a minimizer of $R_{FSP}$ on $\mathbb{H}_{\boldsymbol{\Sigma}} \cap \mathbb{F}$.*

We prove Propositions 1 and 2 in Appendix A.2. To provide some intuition about the mode of convergence in Proposition 2, we point out that $\mathbb{H}_{\boldsymbol{\Sigma}}$-weak convergence of the subsequence $\{\boldsymbol{f}_{n_k}^{\star}\}_{k\in\mathbb{N}}$ implies both (strong/norm) convergence in the path space $\mathbb{B}$ of the Gaussian process $\mathbf{f}$ (i.e., $\lim_{k\to\infty}\|\boldsymbol{f}_{n_k}^{\star} - \boldsymbol{f}^{\star}\|_{\mathbb{B}} = 0$) and pointwise convergence (i.e., $\lim_{k\to\infty}\boldsymbol{f}_{n_k}^{\star}(\boldsymbol{x}) = \boldsymbol{f}^{\star}(\boldsymbol{x})\ \forall \boldsymbol{x} \in \mathbb{X}$).

Finally, under certain technical assumptions, Theorem 4.4 from Cockayne et al. [22] can be used to show that, as $\lambda \to 0$, $P_{\mathbb{B}}^{\boldsymbol{Y},\lambda}$ converges[2] to $P_{\mathbb{F}}^{\boldsymbol{Y}}$ defined by the Radon-Nikodym derivative $P_{\mathbb{F}}^{\boldsymbol{Y}}(\mathrm{d}\boldsymbol{f}) \propto \exp\left(-\Phi^{\boldsymbol{Y}}(\mathrm{d}\boldsymbol{f})\right) P_{\mathbb{F}}(\mathrm{d}\boldsymbol{f})$ where $P_{\mathbb{F}}$ is the regular conditional probability measure $P_{\mathbb{B}}(\,\cdot\mid\mathbf{f} \in \mathbb{F})$.

---

[2] w.r.t. an integral probability metric (IPM)

Summarizing informally, we address the nonexistence of a density-based MAP estimator by constructing a family $\{\boldsymbol{f}_\lambda^\star\}_{\lambda>0}$ of weak modes of the related "relaxed" posteriors $\mathrm{P}_{\mathbb{B}}^{\boldsymbol{Y},\lambda}(\mathrm{d}\boldsymbol{f}) = \mathrm{P}_{\mathbb{B}}^{\boldsymbol{Y}}(\mathrm{d}\boldsymbol{f} \mid d_{\mathbb{B}}(\mathbf{f},\mathbb{F})+\boldsymbol{\epsilon}_\lambda = 0)$ (Proposition 1). These weak modes converge to minimizers $\boldsymbol{f}^\star$ of the optimization problem in Equation (3.3) as $\lambda \to 0$ (Proposition 2). Moreover, under certain technical assumptions, the relaxed posteriors $\mathrm{P}_{\mathbb{B}}^{\boldsymbol{Y},\lambda}$ converge to the "true" posterior $\mathrm{P}_{\mathbb{F}}^{\boldsymbol{Y}}$ as $\lambda \to 0$. Given the above, we conjecture that the $\boldsymbol{f}^\star$ are weak modes of $\mathrm{P}_{\mathbb{F}}^{\boldsymbol{Y}}$, and leave the proof for future work. This motivates using the objective in Equation (3.3) to find the weak mode of the posterior measure $\mathrm{P}_{\mathbb{F}}^{\boldsymbol{Y}}$ that we wish to Laplace-approximate. The next paragraph shows how this objective becomes a valid log-density.

**The FSP-LAPLACE objective as an unnormalized log-density.** As a first step, we use Algorithm 1, discussed in detail below, to train the neural network using the objective function[3]

$$R_{\mathrm{FSP}}(\boldsymbol{w}) := R_{\mathrm{FSP}}(\boldsymbol{f}(\,\cdot\,,\boldsymbol{w})) = -\sum_{i=1}^n \log p(\boldsymbol{y}^{(i)} \mid \boldsymbol{f}(\boldsymbol{x}^{(i)},\boldsymbol{w})) + \frac{1}{2}\|\boldsymbol{f}(\,\cdot\,,\boldsymbol{w}) - \boldsymbol{\mu}\|_{\mathbb{H}_{\boldsymbol{\Sigma}}}^2. \qquad (3.5)$$

We then intuitively want to use the same objective function to compute an approximate posterior over the weights using a Laplace approximation. For this to be well-defined, $R_{\mathrm{FSP}}(\boldsymbol{w})$ needs to be a valid unnormalized negative log-density, i.e., $\boldsymbol{w} \mapsto \exp(-R_{\mathrm{FSP}}(\boldsymbol{w}))$ needs to be integrable. However, without weight-space regularization, this often fails to be the case (e.g., due to continuous symmetries in weight space). Our method works around this issue by linearizing the network locally around $\boldsymbol{w}^\star$ after training (see Equation (2.2)) and then applying a Laplace approximation to $R_{\mathrm{FSP}}^{\mathrm{lin}}(\boldsymbol{w}) := R_{\mathrm{FSP}}(\boldsymbol{f}^{\mathrm{lin}}(\,\cdot\,,\boldsymbol{w}))$. In this case, the RKHS regularizer in $R_{\mathrm{FSP}}^{\mathrm{lin}}$ can be rewritten as

$$\frac{1}{2}\|\boldsymbol{f}^{\mathrm{lin}}(\,\cdot\,,\boldsymbol{w}) - \boldsymbol{\mu}\|_{\mathbb{H}_{\boldsymbol{\Sigma}}}^2 = \frac{1}{2}(\boldsymbol{w} - \boldsymbol{\mu}_{\boldsymbol{w}^\star})^\top \boldsymbol{\Sigma}_{\boldsymbol{w}^\star}^\dagger (\boldsymbol{w} - \boldsymbol{\mu}_{\boldsymbol{w}^\star}) + \text{const.}, \qquad (3.6)$$

where $(\boldsymbol{\Sigma}_{\boldsymbol{w}^\star}^\dagger)_{ij} := \langle (\boldsymbol{J}_{\boldsymbol{w}^\star})_i, (\boldsymbol{J}_{\boldsymbol{w}^\star})_j \rangle_{\mathbb{H}_{\boldsymbol{\Sigma}}}$ (here, $\boldsymbol{\Sigma}_{\boldsymbol{w}^\star}^\dagger$ is the Moore–Penrose pseudoinverse), $\boldsymbol{v}_i := \langle (\boldsymbol{J}_{\boldsymbol{w}^\star})_i, \boldsymbol{f}(\,\cdot\,,\boldsymbol{w}^\star) - \boldsymbol{\mu} \rangle_{\mathbb{H}_{\boldsymbol{\Sigma}}}$, $\boldsymbol{\mu}_{\boldsymbol{w}^\star} := \boldsymbol{w}^\star - \boldsymbol{\Sigma}_{\boldsymbol{w}^\star}\boldsymbol{v}$. Crucially, $\boldsymbol{\Sigma}_{\boldsymbol{w}^\star}$ is positive-(semi)definite. This means that $\exp(-R_{\mathrm{FSP}}^{\mathrm{lin}}(\,\cdot\,))$ is normalizable over $\mathrm{im}(\boldsymbol{\Sigma}_{\boldsymbol{w}^\star})$, i.e., we don't integrate over the null space. Note that this approximation also induces a (potentially degenerate) Gaussian "prior" $\mathbf{w} \sim \mathcal{N}(\boldsymbol{\mu}_{\boldsymbol{w}^\star}, \boldsymbol{\Sigma}_{\boldsymbol{w}^\star})$ over the weights.

Using model linearization, we can also establish a correspondence between the Gaussian process prior in function space and the induced prior over the weights. Namely, the resulting expression is the Lebesgue density of the $\mathbb{H}_{\boldsymbol{\Sigma}}$-orthogonal projection of the GP prior onto the finite-dimensional subspace spanned by the "feature functions" $(\boldsymbol{J}_{\boldsymbol{w}^\star})_i$ learned by the neural network. Hence, our model inherits the prior structure in function space on the features induced by the Jacobian, zeroing out the probability mass in the remaining directions.

## 3.2 Algorithmic Considerations

**Training with the FSP-LAPLACE objective function.** Evaluating the FSP-LAPLACE objective proposed in the previous section requires computing the RKHS norm of the neural network. Unfortunately, this does not admit a closed-form expression in general. Hence, we use the approximation

$$\|\boldsymbol{f}(\,\cdot\,,\boldsymbol{w}) - \boldsymbol{\mu}\|_{\mathbb{H}_{\boldsymbol{\Sigma}}}^2 \approx \underbrace{(\boldsymbol{f}(\boldsymbol{C},\boldsymbol{w}) - \boldsymbol{\mu}(\boldsymbol{C}))^\top \boldsymbol{\Sigma}(\boldsymbol{C},\boldsymbol{C})^{-1}(\boldsymbol{f}(\boldsymbol{C},\boldsymbol{w}) - \boldsymbol{\mu}(\boldsymbol{C}))}_{=\|\boldsymbol{h}_{\boldsymbol{C}}(\,\cdot\,,\boldsymbol{w})\|_{\mathbb{H}_{\boldsymbol{\Sigma}}}^2}, \qquad (3.7)$$

where $\boldsymbol{C} \in \mathbb{X}^{n_C}$ is a set of $n_C$ *context points* and $\boldsymbol{h}_{\boldsymbol{C}}(\boldsymbol{x},\boldsymbol{w}) := \boldsymbol{\Sigma}(\boldsymbol{x},\boldsymbol{C})\boldsymbol{\Sigma}(\boldsymbol{C},\boldsymbol{C})^{-1}(\boldsymbol{f}(\boldsymbol{C},\boldsymbol{w}) - \boldsymbol{\mu}(\boldsymbol{C})) \in \mathbb{H}_{\boldsymbol{\Sigma}}$. The function $\boldsymbol{h}_{\boldsymbol{C}}$ is the minimum-norm interpolant of $\boldsymbol{f}(\,\cdot\,,\boldsymbol{w})-\boldsymbol{\mu}$ at $\boldsymbol{C}$ in $\mathbb{H}_{\boldsymbol{\Sigma}}$. Hence, the estimator of the RKHS norm provably underestimates, i.e., $\|\boldsymbol{h}_{\boldsymbol{C}}(\,\cdot\,,\boldsymbol{w})\|_{\mathbb{H}_{\boldsymbol{\Sigma}}}^2 \leq \|\boldsymbol{f}(\,\cdot\,,\boldsymbol{w}) - \boldsymbol{\mu}\|_{\mathbb{H}_{\boldsymbol{\Sigma}}}^2$.

During training, we need to compute $\|\boldsymbol{h}_{\boldsymbol{C}}(\,\cdot\,,\boldsymbol{w})\|_{\mathbb{H}_{\boldsymbol{\Sigma}}}^2$ at every optimizer step. Since this involves solving a linear system in $n_C$ unknowns and, more importantly, evaluating the neural network on the $n_C$ context points, we need to keep $n_C$ small for computational efficiency. We find that sampling an i.i.d. set of context points at every training iteration from a distribution $\mathrm{P}_{\boldsymbol{C}}$ is an effective strategy for keeping $n_C$ small while ensuring that the neural network is appropriately regularized. The resulting training procedure is outlined in Algorithm 1.

---

[3]In order to simplify notation, we will assume $\mathbb{F} \subset \mathbb{H}_{\boldsymbol{\Sigma}}$ for the remainder of the main paper.

---

**Algorithm 1** RKHS-regularized model training

---

1: **function** FSP-LAPLACE-TRAIN($\boldsymbol{f}, \boldsymbol{w}, \mathcal{GP}(\boldsymbol{\mu}, \boldsymbol{\Sigma}), \mathrm{P}_{\boldsymbol{C}}, \mathbb{D}, b$)
2:      **for all** minibatch $\mathcal{B} = (\boldsymbol{X}_{\mathcal{B}}, \boldsymbol{Y}_{\mathcal{B}}) \sim \mathbb{D}$ of size $b$ **do**
3:          $R_{\mathrm{FSP}}^{(1)}(\boldsymbol{w}^{(i)}) \leftarrow -\frac{n}{b} \sum_{j=1}^{b} \log p(\boldsymbol{y}_{\mathcal{B}}^{(j)} \mid \boldsymbol{f}(\boldsymbol{x}_{\mathcal{B}}^{(j)}, \boldsymbol{w}))$        ▷ *Negative log-likelihood*
4:          $\boldsymbol{C} \leftarrow (\boldsymbol{c}^{(j)})_{j=1}^{n_C} \overset{\text{i.i.d.}}{\sim} \mathrm{P}_{\boldsymbol{C}}$             ▷ *Sample $n_C$ context points*
5:          $R_{\mathrm{FSP}}^{(2)}(\boldsymbol{w}) \leftarrow \frac{1}{2}(\boldsymbol{f}(\boldsymbol{C}, \boldsymbol{w}) - \boldsymbol{\mu}(\boldsymbol{C}))^{\top} \boldsymbol{\Sigma}(\boldsymbol{C}, \boldsymbol{C})^{-1}(\boldsymbol{f}(\boldsymbol{C}, \boldsymbol{w}) - \boldsymbol{\mu}(\boldsymbol{C}))$    ▷ *Equation* (3.7)
6:          $\boldsymbol{w} \leftarrow \textsc{OptimizerStep}(R_{\mathrm{FSP}}^{(1)} + R_{\mathrm{FSP}}^{(2)}, \boldsymbol{w})$
7:      **return** $\boldsymbol{w}$

---

---

**Algorithm 2** Linearized Laplace approximation with Gaussian process priors

---

1: **function** FSP-LAPLACE($\boldsymbol{f}, \mathcal{GP}(\boldsymbol{\mu}, \boldsymbol{\Sigma}), \boldsymbol{C}, \mathbb{D}, \boldsymbol{w}^{\star}$)
2:      $\boldsymbol{v} \leftarrow \boldsymbol{J}_{\boldsymbol{w}^{\star}}(\boldsymbol{C})\boldsymbol{1}/\|\boldsymbol{J}_{\boldsymbol{w}^{\star}}(\boldsymbol{C})\boldsymbol{1}\|_2$       ▷ *Initial Lanczos vector (using forward-mode autodiff)*
3:      $\boldsymbol{L} \leftarrow \textsc{Lanczos}(\boldsymbol{\Sigma}(\boldsymbol{C}, \boldsymbol{C}), \boldsymbol{v}) \in \mathbb{R}^{n_C d' \times r}$    ▷ $r \ll \min(n_C d', p)$ and $\boldsymbol{L}\boldsymbol{L}^{\top} \approx \boldsymbol{\Sigma}(\boldsymbol{C}, \boldsymbol{C})^{\dagger}$
4:      $\boldsymbol{M} \leftarrow \boldsymbol{J}_{\boldsymbol{w}^{\star}}(\boldsymbol{C})^{\top} \boldsymbol{L} \in \mathbb{R}^{p \times r}$            ▷ *using backward-mode autodiff*
5:      $(\boldsymbol{U}_M, \boldsymbol{D}_M, \dots) \leftarrow \mathrm{svd}(\boldsymbol{M})$        ▷ $\boldsymbol{M}\boldsymbol{M}^{\top} = \boldsymbol{U}_M \boldsymbol{D}_M^2 \boldsymbol{U}_M^{\top}$ where $\boldsymbol{U}_M \in \mathbb{R}^{p \times r}$
6:      $\boldsymbol{A} \leftarrow \boldsymbol{D}_M^2 - \sum_{i=1}^{n} \boldsymbol{U}_M^{\top} \boldsymbol{J}_{\boldsymbol{w}^{\star}}(\boldsymbol{x}^{(i)})^{\top} \boldsymbol{H}_{\boldsymbol{w}^{\star}}^{(i)} \boldsymbol{J}_{\boldsymbol{w}^{\star}}(\boldsymbol{x}^{(i)}) \boldsymbol{U}_M$    ▷ *using forward-mode autodiff*
7:      $(\boldsymbol{U}_A, \boldsymbol{D}_A) \leftarrow \mathrm{eigh}(\boldsymbol{A})$                  ▷ $\boldsymbol{U}_A, \boldsymbol{D}_A \in \mathbb{R}^{r \times r}$
8:      $\boldsymbol{S} \leftarrow \boldsymbol{U}_M \boldsymbol{U}_A \boldsymbol{D}_A^{\dagger/2} \in \mathbb{R}^{p \times r}$                ▷ $\boldsymbol{S}\boldsymbol{S}^{\top} \approx \boldsymbol{\Lambda}^{\dagger}$
9:      Find smallest $k$ such that $\mathrm{diag}\left(\boldsymbol{J}_{\boldsymbol{w}^{\star}}(\boldsymbol{C}_i)\boldsymbol{S}_{:,k:r}(\boldsymbol{S}_{:,k:r})^{\top} \boldsymbol{J}_{\boldsymbol{w}^{\star}}(\boldsymbol{C}_i)^{\top}\right) \leq \mathrm{diag}\,\boldsymbol{\Sigma}(\boldsymbol{C}_i, \boldsymbol{C}_i)$    $\forall i$
10:      **return** $\mathcal{N}\left(\boldsymbol{w}^{\star}, \boldsymbol{S}_{:,k:r}(\boldsymbol{S}_{:,k:r})^{\top}\right)$

---

**Efficient linearized Laplace approximations of the FSP-LAPLACE objective.** Once a minimum $\boldsymbol{w}^{\star}$ of $R_{\mathrm{FSP}}$ is found, we compute a linearized Laplace approximation at $\boldsymbol{w}^{\star}$. The Hessian of $R_{\mathrm{FSP}}$ is then

$$\boldsymbol{\Lambda} = \boldsymbol{\Sigma}_{\boldsymbol{w}^{\star}}^{\dagger} - \sum_{i=1}^{n} \boldsymbol{J}_{\boldsymbol{w}^{\star}}(\boldsymbol{x}^{(i)})^{\top} \boldsymbol{H}_{\boldsymbol{w}^{\star}}^{(i)} \boldsymbol{J}_{\boldsymbol{w}^{\star}}(\boldsymbol{x}^{(i)}). \tag{3.8}$$

Again, the RKHS inner products in the entries of $\boldsymbol{\Sigma}_{\boldsymbol{w}^{\star}}^{\dagger}$ do not admit general closed-form expressions. Hence, we use the same strategy as before to estimate $\boldsymbol{\Sigma}_{\boldsymbol{w}^{\star}}^{\dagger} \approx \boldsymbol{J}_{\boldsymbol{w}^{\star}}(\boldsymbol{C})^{\top} \boldsymbol{\Sigma}(\boldsymbol{C}, \boldsymbol{C})^{-1} \boldsymbol{J}_{\boldsymbol{w}^{\star}}(\boldsymbol{C})$ at another set $\boldsymbol{C}$ of context points. Unlike above, for a Laplace approximation, it is vital to use a large number of context points to capture the prior beliefs well. Luckily, $\boldsymbol{\Sigma}_{\boldsymbol{w}^{\star}}^{\dagger}$ only needs to be computed once, at the end of training. But for large networks and large numbers of context points, it is still infeasible to compute or even represent $\boldsymbol{\Sigma}_{\boldsymbol{w}^{\star}}^{\dagger}$ in memory. To address this problem, we devise an efficient routine for computing (a square root of) the approximate posterior covariance matrix $\boldsymbol{\Lambda}^{\dagger}$, outlined in Algorithm 2. Our method is *matrix-free*, i.e. it never explicitly constructs any big (i.e., $p \times p$, $p \times n_C$, or $n_C \times n_C$) matrices in memory. This allows the method to scale to large models.

We start by computing a rank $r \ll \min(n_C d', p)$ approximation of the (pseudo-)inverted kernel Gram matrix $\boldsymbol{\Sigma}(\boldsymbol{C}, \boldsymbol{C})^{\dagger} \approx \boldsymbol{L}\boldsymbol{L}^{\top}$ using fully-reorthogonalized Lanczos iteration [23, Section 10.1] with an embedded $\boldsymbol{L}\boldsymbol{D}\boldsymbol{L}^{\top}$-factorization [23, Section 11.3.5]. This only needs access to matrix-vector products with $\boldsymbol{\Sigma}(\boldsymbol{C}, \boldsymbol{C})$, which can be implemented efficiently without materializing the matrix in memory. Moreover, kernel Gramians typically exhibit rapid spectral decay [see e.g., 24], which makes the low-rank approximation particularly accurate. The low-rank factors $\boldsymbol{L}$ then yield a rank $r$ approximation of $\boldsymbol{\Sigma}_{\boldsymbol{w}^{\star}}^{\dagger} \approx \boldsymbol{M}\boldsymbol{M}^{\top}$ with $\boldsymbol{M} := \boldsymbol{J}_{\boldsymbol{w}^{\star}}(\boldsymbol{C})^{\top} \boldsymbol{L}$, which is embarrassingly parallel and can be computed using backward-mode autodiff to avoid materializing the network Jacobians in memory. An eigendecomposition of $\boldsymbol{M}\boldsymbol{M}^{\top} = \boldsymbol{U}_M \boldsymbol{D}_M^2 \boldsymbol{U}_M^{\top}$ can be computed in $\mathcal{O}(pr^2)$ time from a thin SVD of $\boldsymbol{M} = \boldsymbol{U}_M \boldsymbol{D}_M \boldsymbol{V}_M^{\top}$. The eigenvectors $\boldsymbol{U}_M$ define an orthogonal projector $\boldsymbol{P} := \boldsymbol{U}_M \boldsymbol{U}_M^{\top}$ onto the range and an orthogonal projector $\boldsymbol{P}_0 := \boldsymbol{I} - \boldsymbol{U}_M \boldsymbol{U}_M^{\top}$ onto the nullspace of $\boldsymbol{M}\boldsymbol{M}^{\top}$. This means that we have the decomposition $\boldsymbol{\Lambda} = \boldsymbol{P}\boldsymbol{\Lambda}\boldsymbol{P}^{\top} + \boldsymbol{P}_0 \boldsymbol{\Lambda} \boldsymbol{P}_0^{\top} = \boldsymbol{U}_M \boldsymbol{A} \boldsymbol{U}_M^{\top} + \boldsymbol{P}_0 \boldsymbol{\Lambda} \boldsymbol{P}_0^{\top}$ with

$$\boldsymbol{A} := \boldsymbol{U}_M^{\top} \boldsymbol{\Lambda} \boldsymbol{U}_M = \boldsymbol{D}_M^2 - \sum_{i=1}^{n} \boldsymbol{U}_M^{\top} \boldsymbol{J}_{\boldsymbol{w}^{\star}}(\boldsymbol{x}^{(i)})^{\top} \boldsymbol{H}_{\boldsymbol{w}^{\star}}^{(i)} \boldsymbol{J}_{\boldsymbol{w}^{\star}}(\boldsymbol{x}^{(i)}) \boldsymbol{U}_M. \tag{3.9}$$

We find that $\boldsymbol{P}_0 \boldsymbol{\Lambda} \boldsymbol{P}_0^\top$ is negligible in practice, and hence, we approximate $\boldsymbol{\Lambda} \approx \boldsymbol{U}_M \boldsymbol{A} \boldsymbol{U}_M^\top$ (see Table C.2).

Finally, by computing an eigendecomposition of $\boldsymbol{A} = \boldsymbol{U}_A \boldsymbol{D}_A \boldsymbol{U}_A^\top \in \mathbb{R}^{r \times r}$ in $\mathcal{O}(r^3)$ time, we obtain an eigendecomposition of $\boldsymbol{\Lambda} \approx \boldsymbol{U}_M \boldsymbol{A} \boldsymbol{U}_M^\top = (\boldsymbol{U}_M \boldsymbol{U}_A) \boldsymbol{D}_A (\boldsymbol{U}_M \boldsymbol{U}_A)^\top$, which can be used to obtain a matrix-free representation of (a square root of) the (pseudo-)inverse of $\boldsymbol{\Lambda}$. Unfortunately, due to numerical imprecision, it is difficult to distinguish between zero eigenvalues of $\boldsymbol{A}$ and those with small positive magnitudes. Since we need to pseudo-invert the matrix to obtain the covariance matrix, this can explode predictive variance. As a remedy, we use a heuristic based on the observation that, in a linear-Gaussian model, the marginal variance of the posterior is always upper-bounded by the marginal variance of the prior. Hence, we impose that $\operatorname{diag}\left(\boldsymbol{J}_{\boldsymbol{w}^\star}(\boldsymbol{C}_i) \boldsymbol{\Lambda}^\dagger \boldsymbol{J}_{\boldsymbol{w}^\star}(\boldsymbol{C}_i)\right) \leq \operatorname{diag} \boldsymbol{\Sigma}(\boldsymbol{C}_i, \boldsymbol{C}_i)$ for all $i = 1, \ldots, n_{\boldsymbol{C}}$ by successively truncating the smallest eigenvalues in $\boldsymbol{D}_A$ until the condition is fulfilled. This turns out to be an effective strategy to combat exploding predictive variance in practice.

**Choice of context points.** Methods for regularizing neural networks in function space rely on a set of context points to evaluate the regularizer [8, 9, 25, 26]. The context points should cover the set of input locations where it is plausible that the model might be evaluated when deployed. Popular strategies include uniform sampling from a bounded subset of the input space [8, 9, 25, 26], from the training data [8], and from additional (possibly unlabeled) datasets [25, 26]. For MAP estimation, we choose a uniform context point distribution $\mathbb{P}_C$ or sample from other datasets for high-dimensional inputs. We compute the posterior covariance using samples $\boldsymbol{C}$ from a low-discrepancy sequence, which effectively cover high-dimensional spaces.

## 4 Experiments

We evaluate FSP-LAPLACE on synthetic and real-world data, demonstrating that our method effectively incorporates beliefs specified through a GP prior; that it improves performance on regression tasks for which we have prior knowledge in the form of a kernel (Mauna Loa and ocean current modeling); and that it shows competitive performance on regression, classification, out-of-distribution detection, and Bayesian optimization tasks compared to baselines.

**Baselines.** We compare FSP-LAPLACE to a deterministic neural network fit using maximum a-posteriori estimation with an isotropic Gaussian prior on the weights (MAP) and a neural network for which we additionally compute the standard linearized Laplace approximation (Laplace) [17]. We further compare our method to FVI [8], which uses GP priors with VI, to a Gaussian process (GP) [13] when the size of the dataset allows it, and to a sparse Gaussian process (sparse GP) [27]. We use the full Laplace approximation when the size of the neural network allows it, and otherwise consider the K-FAC or diagonal approximations [28]. All neural networks share the same architecture.

**Qualitative evaluation on synthetic data.** We consider two synthetic data tasks: a 1-dimensional regression task with randomly drawn noisy measurements of the function $y = \sin(2\pi x)$, and the 2-dimensional two-moons classification task from the `scikit-learn` library [29]. For the regression task, data points are shown as gray circles, functions drawn from the posterior as green lines and the inferred mean function as a red line (see Figures 1 and C.1 to C.4). For the classification task, we plot the predictive mean and 2-standard-deviations of the predictions for class 1 (blue circles) in Figures C.5 and C.6. We apply FSP-LAPLACE to the data with different GP priors and find that it successfully adapts to the specified beliefs. For instance, by varying the GP prior, we can make our method generate functions that are periodic within the support of the context points (Figure C.2), and control their smoothness (Figure 1) and length scale (Figure C.3) without modifying the network's architecture, or adding features. Such flexibility is impossible with the isotropic Gaussian weight-space priors used in the Laplace and MAP baselines. These results carry over to classification, where our method shows a smooth decision boundary when equipped with an RBF kernel (Figure C.6), and a rough decision boundary when equipped with a Matern-1/2 kernel (Figure C.5). Unlike the Laplace and MAP baselines, whose predictions remain confident beyond the support of the data, the FSP-LAPLACE's posterior mean reverts to the zero-mean prior, and its posterior variance increases. Our method also captures the properties specified by the GP prior better than FVI, especially for rougher Matern-1/2 kernels (Figures 1 and C.5). While a sufficient number of context points is necessary to capture the beliefs specified by the GP prior, we find that, even with a very small number

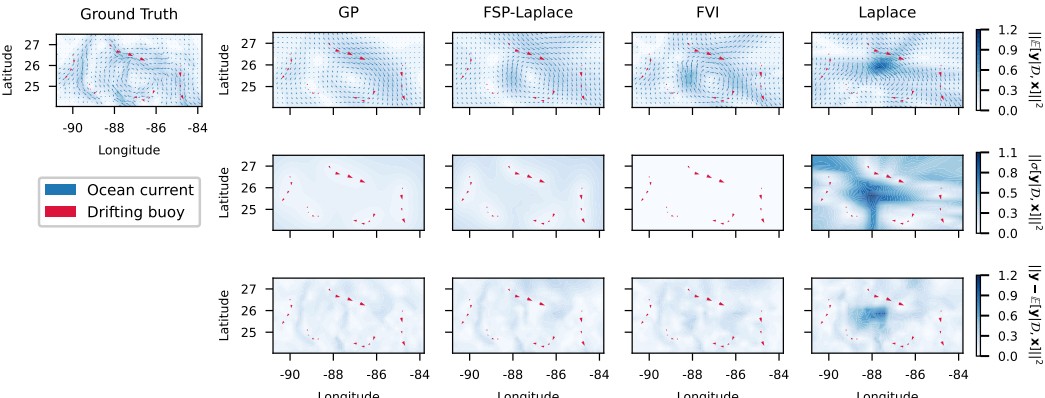

Figure 2: Results for the ocean current modeling experiment. We report the mean velocity vectors, the norm of their standard-deviation and the squared errors of compared methods. Unlike the Laplace, we find that FSP-LAPLACE accurately captures ocean current dynamics.

Table 1: Results for the Mauna Loa $CO_2$ prediction and ocean current modeling tasks. Incorporating knowledge via the GP prior in our FSP-LAPLACE improves performance over the standard Laplace.

| DATASET | EXPECTED LOG-LIKELIHOOD (↑) | | | | MEAN SQUARED ERROR (↓) | | | |
|---|---|---|---|---|---|---|---|---|
| | FSP-LAPLACE (OURS) | FVI | LAPLACE | GP | FSP-LAPLACE (OURS) | FVI | LAPLACE | GP |
| MAUNA LOA | -6'015.39 ± 443.13 | -13'594.57 ± 1'303.26 | -16'139.85 ± 2'433.70 | **-2'123.90 ± 0.00** | **8.90 ± 1.15** | 38.38 ± 2.10 | 28.26 ± 6.87 | 35.66 ± 0.00 |
| OCEAN CURRENT | **0.2823 ± 0.0009** | -35.2741 ± 1.0278 | -126'845.4063 ± 14'470.5270 | -0.5069 ± 0.0000 | 0.0169 ± 0.0001 | 0.0184 ± 0.0003 | 0.0320 ± 0.0016 | **0.0131 ± 0.0000** |

of context points, our method produces useful uncertainty estimates and reverts to the prior mean (Figures C.7 and C.8).

## 4.1 Quantitative evaluation on real-world data

We now move on to investigate FSP-LAPLACE on two real-world scientific modeling tasks: forecasting the concentration of $CO_2$ at the Mauna Loa observatory and predicting ocean currents in the Gulf of Mexico. We then assess the performance of our method on standard benchmark regression, classification, out-of-distribution detection, and Bayesian optimization tasks. When reporting results, we bold the score with the highest mean as well as any scores whose standard-error bars overlap.

**Mauna Loa dataset.** We consider the task of modeling the monthly average atmospheric $CO_2$ concentration at the Mauna Loa observatory in Hawaii from 1974 to 2024 using data collected from the NORA global monitoring laboratory[4] [30]. This 1-dimensional dataset is very accurately modeled by a combination of multiple kernels proposed in Rasmussen and Williams [13, Section 5.4.3]. We equip FSP-LAPLACE with this informative kernel and with additional periodic features, and compare to FVI with the same prior and additional periodic features, to a GP with the same prior, and to the linearized Laplace with the same additional periodic features. Using periodic features (partially) reflects the prior knowledge carried by the kernel. Results are presented by Table 1 and Figure C.9 in the Appendix. We find that incorporating prior beliefs both via an informative prior and periodic features in FSP-LAPLACE significantly reduces the mean squared error (MSE) compared to Laplace and GP baselines, and that our method is also more accurate than FVI, which uses variational inference. In terms of expected log-likelihood, all neural networks under-estimate the likelihood scale, which results in poorer scores than the GP.

**Ocean current modeling.** We further evaluate FSP-LAPLACE's ability to take into account prior knowledge by considering an ocean current modeling task where we are given sparse 2-dimensional observations of velocities of ocean drifter buoys, and we are interested in estimating ocean currents further away from the buoys. For this, we consider the GulfDrifters dataset [31] and we follow the setup by Shalashilin [32]. We incorporate known physical properties of ocean currents into the considered models by applying the Helmholtz decomposition to the GP prior's kernel [33] as well as to the neural network. Results are shown in Table 1 and in Figure 2. We find that FSP-LAPLACE

---

[4]https://gml.noaa.gov/ccgg/trends/data.html

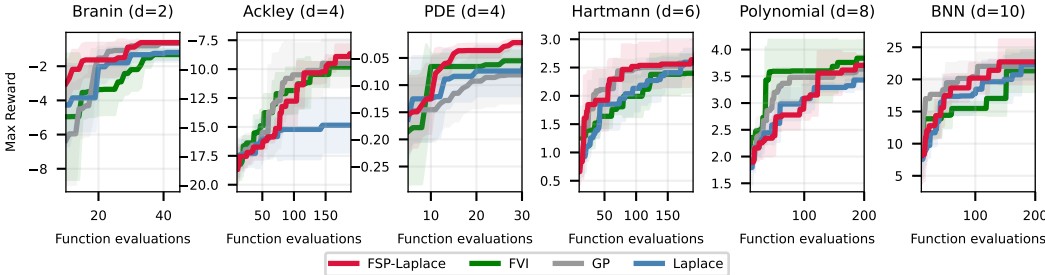

Figure 3: Results using our method (FSP-LAPLACE) as a surrogate model for Bayesian optimization. We find that FSP-LAPLACE performs particularly well on lower-dimensional problems, where it converges more quickly and to higher rewards than the Laplace, obtaining comparable scores as the Gaussian process (GP).

Table 2: Test expected log-likelihood, accuracy, expected calibration error and OOD detection accuracy on MNIST and FashionMNIST. FSP-LAPLACE performs strongly among baselines matching the accuracy of best-performing baselines and obtaining the highest expected log-likelihood and out-of-distribution detection accuracy.

| DATASET | METRIC | FSP-LAPLACE (KMNIST) | FSP-LAPLACE (RND) | FVI (KMNIST) | FVI (RND) | LAPLACE | MAP | SPARSE GP |
|---------|--------|----------------------|-------------------|--------------|-----------|---------|-----|-----------|
| MNIST | LOG-LIKELIHOOD (↑) | -0.043 ± 0.001 | **-0.037 ± 0.001** | -0.238 ± 0.006 | -0.145 ± 0.005 | -0.043 ± 0.001 | -0.039 ± 0.001 | -0.301 ± 0.002 |
| | ACCURACY (↑) | **0.989 ± 0.000** | **0.989 ± 0.000** | 0.943 ± 0.001 | 0.976 ± 0.001 | **0.989 ± 0.001** | 0.987 ± 0.000 | 0.930 ± 0.001 |
| | ECE (↓) | **0.002 ± 0.000** | **0.002 ± 0.000** | 0.073 ± 0.003 | 0.064 ± 0.001 | 0.013 ± 0.001 | 0.003 ± 0.000 | 0.035 ± 0.001 |
| | OOD ACCURACY (↑) | **0.977 ± 0.003** | 0.895 ± 0.011 | 0.891 ± 0.006 | 0.894 ± 0.010 | 0.907 ± 0.016 | 0.889 ± 0.009 | 0.904± 0.020 |
| FMNIST | LOG-LIKELIHOOD (↑) | **-0.255 ± 0.002** | -0.259 ± 0.003 | -0.311 ± 0.005 | -0.300 ± 0.002 | -0.290 ± 0.006 | -0.281 ± 0.002 | -0.479 ± 0.002 |
| | ACCURACY (↑) | **0.909 ± 0.001** | **0.909 ± 0.001** | 0.906 ± 0.002 | **0.910 ± 0.002** | 0.897 ± 0.002 | 0.900 ± 0.001 | 0.848 ± 0.001 |
| | ECE (↓) | 0.018 ± 0.002 | 0.020 ± 0.003 | 0.024 ± 0.002 | 0.027 ± 0.005 | **0.014 ± 0.002** | 0.016 ± 0.002 | 0.027 ± 0.001 |
| | OOD ACCURACY (↑) | **0.994 ± 0.001** | 0.797 ± 0.007 | 0.975 ± 0.002 | 0.925 ± 0.005 | 0.801 ± 0.014 | 0.794 ± 0.010 | 0.938 ± 0.007 |

strongly improves over Laplace, FVI and GP in terms of expected log-likelihood, and performs competitively in terms of mean squared error (MSE). FSP-LAPLACE also improves MSE and test expected log-likelihood over FVI, which strongly underestimates the predictive variance.

**Image classification.** We further evaluate our method on the MNIST [34] and FashionMNIST [35] image classification datasets using a convolutional neural network. We compare our model to FVI, Laplace, MAP, and Sparse GP baselines, as the scale of the datasets forbids exact GP inference. For FSP-LAPLACE and FVI, we compare using context points drawn from a uniform distribution (RND) and drawn from the Kuzushiji-MNIST dataset (KMNIST) following the setup by Rudner et al. [25]. Results are presented in Table 2. Although these datasets are particularly challenging to our method, which is regularized in function space, we find that FSP-LAPLACE performs strongly and matches or exceeds the expected log-likelihood and predictive accuracy of best-performing baselines. It also yields well-calibrated models with low expected calibration error (ECE).

**Out-of-distribution detection.** We now investigate whether the epistemic uncertainty of FSP-LAPLACE is predictive for out-of-distribution detection (OOD). We follow the setup by Osawa et al. [36] and report the accuracy of a single threshold to classify OOD from in-distribution (ID) data based on the predictive uncertainty. Additional details are provided in Appendix B.2. FSP-LAPLACE with context points sampled from KMNIST performs strongly, obtaining the highest out-of-distribution detection accuracy (see Table 2, note that FSP-LAPLACE makes no assumption on whether context points are in or out of distribution). This can be further observed in Figure C.10 in the Appendix, where the predictive entropy of ID data points is tightly peaked around 0, whereas the predictive entropy of OOD data points is highly concentrated around the maximum entropy of the softmax distribution ($\ln(10) \approx 2.3$). With RND context points, our method performs comparably to the Laplace baseline.

**Bayesian optimization.** We finally evaluate the epistemic uncertainty of FSP-LAPLACE as a surrogate model for Bayesian optimization. We consider a setup derived from Li et al. [7], comparing to FVI, the linearized Laplace, and to a GP. Results are summarized in Figure 3. We find that our method performs particularly well on lower-dimensional tasks, where it converges both faster and to higher rewards than Laplace, and noticeably strongly improves over a Gaussian process on PDE. On higher-dimensional tasks, our method performs comparatively well to Laplace and GP baselines.

# 5 Related work

**Laplace approximation in neural networks.** First introduced by MacKay [37], the Laplace approximation gained strong traction in the Bayesian deep learning community with the introduction of scalable log-posterior Hessian approximations [28, 38], and the so-called linearized Laplace, which solves the underfitting issue observed with standard Laplace [17, 39–41]. In addition to epistemic uncertainty estimates, the Laplace approximation also provides a method to select prior parameters via marginal likelihood estimation [11, 12]. Recent work has made the linearized Laplace more scalable by restricting inference to a subset of parameters [42], by exploiting its GP formulation [17] to apply methods from the scalable GP literature [43–45], or by directly sampling from the Laplace approximation without explicitly computing the covariance matrix [46]. While these approaches use the GP formulation to make the linearized Laplace more scalable, we are unaware of any method that uses GP priors to incorporate interpretable prior beliefs within the Laplace approximation.

**BNNs with function-space priors.** In the context of variational inference, function-space priors in BNNs demonstrate improvements in predictive performance compared to weight-space priors [8]. While this idea might seem sound, it turns out that the KL divergence in the VI objective is infinite for most cases of interest due to mismatching supports between the function-space prior and BNN's predictive posterior [14], which therefore requires additional regularization to be well defined [8]. Due to this issue, other work [47] considers generalized VI [10] using the regularized KL divergence [48] or abandons approximating the neural network's posterior and instead uses deterministic neural networks as basis functions for Bayesian linear regression [15] or for the mean of a sparse GP [49]. In contrast, our method does not compute a divergence in function space, but only the RKHS norm under the prior's kernel, thus circumventing the issue of mismatching support. Alternatively, rather than directly placing a prior on the function generated by a BNN, researchers have investigated methods to find weight-space priors whose pushforward approximates a target function-space measure by minimizing a divergence [9, 50], using the Ridgelet transform [51], or changing the BNN's architecture [52].

**Regularizing neural networks in function space.** Arguing that one ultimately only cares about the output function of the neural network, it has been proposed to regularize neural networks in function space, both showing that norms could be efficiently estimated and that such regularization schemes performed well in practice [26, 53–55]. Unlike FSP-LAPLACE, none of these methods allow to specify informative beliefs via a GP prior. Our method uses the same RKHS norm estimator as Chen et al. [54] (however, under a different kernel) by sampling a new batch of context points at each update step. Similarly, Rudner et al. [26] propose an empirical prior on the weights that regularizes the neural network in function space, which they use for MAP estimation or approximate posterior inference. The MAP objective resembles ours, but unlike our method, uses the kernel induced by the last layer of the neural network and includes an additional Gaussian prior on the weights.

# 6 Conclusion

We propose a method for applying the Laplace approximation to neural networks with interpretable Gaussian process priors in function space. This addresses the issue that conventional applications of approximate Bayesian inference methods to neural networks require posing a prior in weight space, which is virtually impossible because weight space is not interpretable. We address the non-existence of densities in (infinite-dimensional) function space by generalizing the notion of a MAP estimate to the limit of a sequence of weak modes of related posterior measures, leading us to propose a simple objective function. We further mitigate the computational cost of calculating high-dimensional curvature matrices using scalable methods from matrix-free linear algebra. By design, our method works best in application domains where prior information can be encoded in the language of Gaussian processes. This is confirmed by experiments on scientific data and Bayesian optimization. In high-dimensional spaces, where explicit prior knowledge is difficult to state, Gaussian process priors are naturally at a disadvantage. While we do demonstrate superior performance on image data, it is yet unclear how to find good function-space priors in such high-dimensional spaces.

## Acknowledgments and Disclosure of Funding

TC, MP, PH, and RB are funded by the DFG Cluster of Excellence "Machine Learning - New Perspectives for Science", EXC 2064/1, project number 390727645; the German Federal Ministry of Education and Research (BMBF) through the Tübingen AI Center (FKZ: 01IS18039A); and funds from the Ministry of Science, Research and Arts of the State of Baden-Württemberg. MP and PH gratefully acknowledge financial support by the European Research Council through ERC StG Action 757275 / PANAMA, and ERC CoG Action 101123955 ANUBIS. VF was supported by a Branco Weiss Fellowship. RB further acknowledges funding by the German Research Foundation (DFG) for project 448588364 of the Emmy Noether Programme. The authors thank the International Max Planck Research School for Intelligent Systems (IMPRS-IS) for supporting TC and MP.

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

# Appendix

## A  Theory

### A.1  Assumptions and their applicability

**Assumption A.1.** $\mathbf{f} \sim \mathcal{GP}(\boldsymbol{\mu}, \boldsymbol{\Sigma})$ is a $d'$-output Gaussian process with index set $\mathbb{X}$ on $(\Omega, \mathcal{A}, \mathrm{P})$ such that

- (i) the paths of $\mathbf{f}$ lie (P-almost surely) in a real separable Banach space $\mathbb{B}$ of $\mathbb{R}^{d'}$-valued functions on $\mathbb{X}$ with continuous point evaluation maps $\delta_{\boldsymbol{x}} \colon \mathbb{B} \to \mathbb{R}^{d'}$, and
- (ii) $\omega \mapsto \mathbf{f}(\cdot, \omega)$ is a Gaussian random variable with values in $(\mathbb{B}, \mathcal{B}(\mathbb{B}))$.

We denote the law of $\omega \mapsto \mathbf{f}(\cdot, \omega)$ by $\mathrm{P}_{\mathbb{B}}$.

For this paper, we focus on $\mathbb{B} = C(\mathbb{X})$, with $\mathbb{X}$ being a compact metric space. In this case, Assumption A.1(i) can be verified from regularity properties of the prior covariance function $\boldsymbol{\Sigma}$ [see, e.g., 56]. Moreover, the sufficient criteria from Pförtner et al. [19, Section B.2] show that Assumption A.1(ii) also holds in this case.

**Assumption A.2.** The potential $\Phi^{\boldsymbol{Y}} \colon \mathbb{B} \to \mathbb{R}$ is (norm-)continuous and, for each $\eta > 0$, there is $K(\eta) \in \mathbb{R}$ such that $\Phi^{\boldsymbol{Y}}(\boldsymbol{f}) \geq K(\eta) - \eta \|\boldsymbol{f}\|_{\mathbb{B}}^2$ for all $\boldsymbol{f} \in \mathbb{B}$.

This holds if the negative log-likelihood functions $\ell^{(i)} \colon \mathbb{R}^{d'} \to \mathbb{R}$, $\boldsymbol{f}^{(i)} \mapsto -\log p(\boldsymbol{y}^{(i)} \mid \boldsymbol{f}^{(i)})$ are continuous and, for all $\eta > 0$, there is $K(\eta) \in \mathbb{R}$ such that $\ell^{(i)}(\boldsymbol{f}^{(i)}) > K(\boldsymbol{f}^{(i)}) + \eta \|\boldsymbol{f}^{(i)}\|^2$ for all $\boldsymbol{f}^{(i)} \in \mathbb{R}^{d'}$. For instance, this is true for a Gaussian likelihood $\ell^{(i)}(\boldsymbol{f}^{(i)}) = \frac{1}{2\lambda_i^2} \|\boldsymbol{y}^{(i)} - \boldsymbol{f}^{(i)}\|_2^2$.

**Assumption A.3.** (i) $\mathbb{H}_{\boldsymbol{\Sigma}} \cap \mathbb{F}$ is nonempty, (ii) $\mathbb{H}_{\boldsymbol{\Sigma}} \cap \mathbb{F}$ is closed in $\mathbb{B} \supset \mathbb{H}_{\boldsymbol{\Sigma}}$, and (iii) $\mathbb{H}_{\boldsymbol{\Sigma}} \cap \mathbb{F} \subset \mathbb{B}$ has the Heine-Borel property, i.e., all closed and bounded subsets of $\mathbb{H}_{\boldsymbol{\Sigma}} \cap \mathbb{F}$ are compact in $\mathbb{B}$.

Assumption A.3(i) can be verified using a plethora of results from RKHS theory. For instance, for Sobolev kernels like the Matérn family used in the experiments, the RKHS is norm-equivalent to a Sobolev space [see, e.g., 57]. In this case, we only need the NN to be sufficiently often (weakly) differentiable on the interior of its compact domain $\mathbb{X}$. The closure property from Assumption A.3(ii) is more difficult to verify directly. However, if we assume that $\mathbb{W}$ is compact and that the map $\mathbb{W} \to \mathbb{B}$, $\boldsymbol{w} \mapsto \boldsymbol{f}(\cdot, \boldsymbol{w})$ is continuous, then $\mathbb{F}$ is compact (and hence closed) as the image of a compact set under a continuous function. This is a reasonable assumption, since, in practice, the weights of a neural network are represented as machine numbers with a maximal magnitude, meaning that $\mathbb{W}$ is always contained in an $\ell_\infty$ ball of fixed radius. Incidentally, compactness of $\mathbb{F}$ also entails the Heine-Borel property from Assumption A.3(iii). Alternatively, $\mathbb{F}$ also has the Heine-Borel property if it is a topological manifold (e.g., a Banach manifold), since it is necessarily finite-dimensional.

### A.2  Proofs

**Lemma A.1.** *Let $(\mathbb{X}, d)$ be a metric space, $\mathbb{A} \subseteq \mathbb{X}$ nonempty, and*

$$d(\cdot, \mathbb{A}) \colon \mathbb{X} \to \mathbb{R}_{\geq 0}, x \mapsto \inf_{a \in \mathbb{A}} d(x, a).$$

*Then $d(\cdot, \mathbb{A})$ is 1-Lipschitz.*

*Proof.* For all $x_1, x_2 \in \mathbb{X}$ we have

$$
\begin{aligned}
d(x_2, \mathbb{A}) &= \inf_{a \in \mathbb{A}} d(x_2, a) \\
&\leq d(x_2, x_1) + \inf_{a \in \mathbb{A}} d(x_1, a) \\
&= d(x_1, x_2) + d(x_1, \mathbb{A})
\end{aligned}
$$

by the triangle inequality and hence $d(x_2, \mathbb{A}) - d(x_1, \mathbb{A}) \leq d(x_1, x_2)$. Since this argument is symmetric in $x_1$ and $x_2$, this also shows that

$$-(d(x_2, \mathbb{A}) - d(x_1, \mathbb{A})) = d(x_1, \mathbb{A}) - d(x_2, \mathbb{A}) \leq d(x_2, x_1) = d(x_1, x_2).$$

All in all, we obtain $|d(x_2, \mathbb{A}) - d(x_1, \mathbb{A})| \leq d(x_1, x_2)$.  □

**Lemma A.2.** *Let $(\mathbb{X}, d)$ be a metric space and $\mathbb{A} \subseteq \mathbb{X}$ a closed, nonempty subset with the Heine-Borel property. Then $\inf_{a \in \mathbb{A}} d(x, a)$ is attained for all $x \in \mathbb{X}$.*

*Proof.* Let $x \in \mathbb{X}$ and $r > r_x := \inf_{a \in \mathbb{A}} d(x, a)$. Then $\mathbb{A} \cap \bar{B}_r(x) \neq \emptyset$ as well as $d(x, a) > r$ for all $a \in \mathbb{A} \setminus \bar{B}_r(x)$ and thus $\inf_{a \in \mathbb{A}} d(x, a) = \inf_{a \in \mathbb{A} \cap \bar{B}_r(x)} d(x, a)$. Moreover, $\mathbb{A} \cap \bar{B}_r(x)$ is compact by the Heine-Borel property and $d(x, \cdot)$ is continuous. Hence, the claim follows from the Weierstrass extreme value theorem. $\square$

**Proposition 1.** *Let Assumptions A.1 to A.3 hold. For $\lambda > 0$, define $\Phi^{Y,\lambda} \colon \mathbb{B} \to \mathbb{R}$, $f \mapsto \Phi^Y(f) + \frac{1}{2\lambda^2} d_{\mathbb{B}}^2(f, \mathbb{F})$. Then the posterior measure $P_{\mathbb{B}}^{Y,\lambda}(df) \propto \exp\left(-\Phi^{Y,\lambda}(df)\right) P_{\mathbb{B}}(df)$ has at least one weak mode $f^\star \in \mathbb{H}_\Sigma$, and the weak modes of $P_{\mathbb{B}}^{Y,\lambda}$ coincide with the minimizers of*

$$R_{FSP}^\lambda \colon \mathbb{H}_\Sigma \to \mathbb{R}, \, f \mapsto \Phi^{Y,\lambda}(f) + \frac{1}{2}\|f - \mu\|_{\mathbb{H}_\Sigma}^2. \tag{3.4}$$

*Proof.* $d_{\mathbb{B}}(\cdot, \mathbb{F})$ is (globally) 1-Lipschitz by Lemma A.1 and bounded from below by 0. Hence, $\Phi^{Y,\lambda} = \Phi^Y + \frac{1}{2\lambda^2} d_{\mathbb{B}}^2(\cdot, \mathbb{F})$ is continuous and for all $\eta > 0$, there is $K(\eta) \in \mathbb{R}$ such that

$$\Phi^{Y,\lambda}(f) \geq K(\eta) - \eta\|f\|_{\mathbb{B}}^2 + \underbrace{\frac{1}{2\lambda^2} d_{\mathbb{B}}^2(f, \mathbb{F})}_{\geq 0} \geq K(\eta) - \eta\|f\|_{\mathbb{B}}^2$$

by Assumption A.2. The statement then follows from Theorem 1.1 in Lambley [20]. $\square$

**Proposition 2.** *Let Assumptions A.1 to A.3 hold. Let $\{\lambda_n\}_{n \in \mathbb{N}} \subset \mathbb{R}_{>0}$ with $\lambda_n \to 0$, and $\{f_n^\star\}_{n \in \mathbb{N}} \subset \mathbb{H}_\Sigma$ such that $f_n^\star$ is a minimizer of $R_{FSP}^{\lambda_n}$. Then $\{f_n^\star\}_{n \in \mathbb{N}}$ has an $\mathbb{H}_\Sigma$-weakly convergent subsequence with limit $f^\star \in \mathbb{H}_\Sigma \cap \mathbb{F}$. Moreover, $f^\star$ is a minimizer of $R_{FSP}$ on $\mathbb{H}_\Sigma \cap \mathbb{F}$.*

Our proof makes use of ideas from Dashti et al. [58] and Lambley [20].

*Proof.* Without loss of generality, we assume $\mu = 0$. By Assumption A.2, there are constants $K, \alpha > 0$ such that $R_{\mathrm{FSP}}(f) \geq K + \alpha\|f\|_{\mathbb{H}_\Sigma}^2$ for all $f \in \mathbb{H}_\Sigma$ [20, Section 4.1]. Now fix an arbitrary $f \in \mathbb{H}_\Sigma \cap \mathbb{F}$. Then

$$
\begin{aligned}
R_{\mathrm{FSP}}(f) &= R_{\mathrm{FSP}}^{\lambda_n}(f) \\
&\geq R_{\mathrm{FSP}}^{\lambda_n}(f_n^\star) \\
&= R_{\mathrm{FSP}}(f_n^\star) + \frac{1}{2\lambda_n^2} d_{\mathbb{B}}^2(f_n^\star, \mathbb{F}) & \text{(A.1)} \\
&\geq K + \alpha\|f_n^\star\|_{\mathbb{H}_\Sigma}^2 + \frac{1}{2\lambda_n^2} d_{\mathbb{B}}^2(f_n^\star, \mathbb{F}) & \text{(A.2)} \\
&\geq K + \alpha\|f_n^\star\|_{\mathbb{H}_\Sigma}^2,
\end{aligned}
$$

and hence

$$\|f_n^\star\|_{\mathbb{H}_\Sigma}^2 \leq \frac{1}{\alpha}\left(R_{\mathrm{FSP}}(f) - K\right),$$

i.e. the sequence $\{f_n^\star\}_{n \in \mathbb{N}} \subset \mathbb{H}_\Sigma$ is bounded. By the Banach-Alaoglu theorem [59, Theorems V.3.1 and V.4.2(d)] and the Eberlein-Šmulian theorem [59, Theorem V.13.1], there is a weakly convergent subsequence $\{f_{n_k}^\star\}_{k \in \mathbb{N}}$ with limit $f^\star \in \mathbb{H}_\Sigma$.

We need to show that $f^\star \in \mathbb{F}$. From Equation (A.2), it follows that

$$0 \leq d_{\mathbb{B}}(f_{n_k}^\star, \mathbb{F}) \leq \lambda_{n_k} \sqrt{2(R_{\mathrm{FSP}}(f) - K - \alpha\|f_{n_k}^\star\|_{\mathbb{H}_\Sigma}^2)} \leq \lambda_{n_k} \sqrt{2(R_{\mathrm{FSP}}(f) - K)},$$

where the right-hand side converges to 0 as $k \to \infty$. Hence, $\lim_{k \to \infty} d_{\mathbb{B}}(f_{n_k}^\star, \mathbb{F}) = 0$. The embedding $\iota \colon \mathbb{H}_\Sigma \to \mathbb{B}$ is compact [60, Corollary 3.2.4] and, by Lemma A.1, $d_{\mathbb{B}}(\cdot, \mathbb{F}) \colon \mathbb{B} \to \mathbb{R}$ is continuous, which implies that $d_{\mathbb{B}}(\iota[\cdot], \mathbb{F}) \colon \mathbb{H}_\Sigma \to \mathbb{R}$ is sequentially weakly continuous. Hence, $d_{\mathbb{B}}(f^\star, \mathbb{F}) = \lim_{k \to \infty} d_{\mathbb{B}}(f_{n_k}^\star, \mathbb{F}) = 0$ and, by Assumption A.3 and Lemma A.2, it follows that $f^\star \in \mathbb{F}$.

Finally, we show that $\boldsymbol{f}^\star$ is a minimizer of $R_{\text{FSP}}$ on $\mathbb{H}_{\boldsymbol{\Sigma}} \cap \mathbb{F}$. Since $\Phi^{\boldsymbol{Y}} \colon \mathbb{B} \to \mathbb{R}$ is continuous and $\iota$ is compact, $\Phi^{\boldsymbol{Y}} \circ \iota \colon \mathbb{H}_{\boldsymbol{\Sigma}} \to \mathbb{R}$ is sequentially weakly continuous. Moreover, we have $\limsup_{k\to\infty} \|\boldsymbol{f}^\star_{n_k}\|^2_{\mathbb{H}_{\boldsymbol{\Sigma}}} \geq \|\boldsymbol{f}^\star\|^2_{\mathbb{H}_{\boldsymbol{\Sigma}}}$, since Hilbert norms are sequentially weakly lower-semicontinuous. Hence, $\limsup_{k\to\infty} R_{\text{FSP}}(\boldsymbol{f}^\star_{n_k}) \geq R_{\text{FSP}}(\boldsymbol{f}^\star)$. Now, by Equation (A.1),

$$R_{\text{FSP}}(\boldsymbol{f}) \geq R_{\text{FSP}}(\boldsymbol{f}^\star) + \limsup_{k\to\infty} \frac{1}{2\lambda^2_{n_k}} d^2_{\mathbb{B}}(\boldsymbol{f}^\star_{n_k}, \mathbb{F}),$$

for all $\boldsymbol{f} \in \mathbb{H}_{\boldsymbol{\Sigma}} \cap \mathbb{F}$. For $\boldsymbol{f} = \boldsymbol{f}^\star$ this implies that $\limsup_{k\to\infty} \frac{1}{2\lambda^2_{n_k}} d^2_{\mathbb{B}}(\boldsymbol{f}^\star_{n_k}, \mathbb{F}) = 0$. All in all, we arrive at $R_{\text{FSP}}(\boldsymbol{f}) \geq R_{\text{FSP}}(\boldsymbol{f}^\star)$ for all $\boldsymbol{f} \in \mathbb{H}_{\boldsymbol{\Sigma}} \cap \mathbb{F}$, i.e. $\boldsymbol{f}^\star$ is a minimizer of $R_{\text{FSP}}$. $\qquad\square$

# B Experimental setup

## B.1 Qualitative experiments with synthetic data

**Regression.** We sample points from the corresponding generative model

$$y_i = \sin(2\pi x_i) + \epsilon \quad \text{with} \quad \epsilon \sim \mathcal{N}\left(0, \sigma_n^2\right) \tag{B.1}$$

using $\sigma_n = 0.1$ and draw $x_i \sim \mathcal{U}([-1, -0.5] \cup [0.5, 1])$. We plot data points as gray circles, functions sampled from the approximate posterior as green lines, the empirical mean function as a red line and its empirical 2-standard-deviation interval around the mean as a green surface. All neural networks have the same two hidden-layer architecture with 50 neurons per layer and hyperbolic tangent ($\tanh$) activation functions. For FSP-LAPLACE, we use 100 context points placed on a regular grid and run a maximum of 500 Lanczos iterations. For FVI, we sample 100 context points drawn from $\mathcal{U}([-2, 2])$ at each update. Except when stated otherwise, we consider a centered GP prior and find the parameters of the covariance function by maximizing the log-marginal likelihood [13]. For the Laplace, we use the full generalized Gauss-Newton matrix and an isotropic Gaussian prior with scale $\sigma_p = 1$. The MAP estimate uses the same prior. We find the parameters of the Gaussian process priors by maximizing the log-marginal likelihood and the parameters of the sparse GP by maximizing the evidence lower bound [13].

**Classification.** We sample randomly perturbed data points from the two moons data [29] with noise level $\sigma_n = 0.1$. We plot the the data points from class 0 as red dots and those from class 1 as blue dots. We show the mean (upper row) and 2-standard-deviation (bottom row) of the probability that a sample $\boldsymbol{x}$ belongs to class 1 under the approximate posterior, which we estimate using $K = 100$ samples. We consider a two hidden-layer neural network with 100 neurons per layer and hyperbolic tangent activation functions. For FSP-LAPLACE, we use 100 context points placed on a regular grid over $[-3.75, 3.75] \times [-3.75, 3.75]$ and limit Lanczos to run for at most 500 iterations. For FVI, we sample 100 context points from $\mathcal{U}([-3.75, 3.75]^2)$ at each update. We consider a centered GP prior and find the parameters of the covariance function by maximizing the log-marginal likelihood [13] using the reparameterization of classifications labels into regression targets from Milios et al. [61]. For the Laplace, we use the full generalized Gauss-Newton matrix and an isotropic Gaussian prior with scale $\sigma_p = 1$. The MAP estimate uses the same prior. For the Gaussian process, we Laplace-approximate the intractable GP posterior and find the prior parameters by maximizing the log-marginal likelihood [13]. Sparse GP parameters are found by maximizing the ELBO [13].

## B.2 Quantitative experiments with real-world data

**Mauna Loa.** We consider the Mauna Loa dataset which tracks the monthly average atmospheric $CO_2$ concentration at the Mauna Loa observatory in Hawaii from 1974 to 2024 [30]. We consider the first $70\%$ of the data in chronological order as the train set (from 1974 to 2005) and the last $30\%$ as the test set (from 2009 to 2024). We standardize the features (time) and regression targets ($CO_2$ concentration). We use two hidden-layer neural networks with hyperbolic tangent activations and 50 units each. We augment the input of the neural networks with an additional sine and cosine transformation of the features i.e., we use the feature vectors $(t_i, \sin(2\pi t_i/T), \cos(2\pi t_i/T))$ where $t_i$ is the time index and $T$ is the period used in Rasmussen and Williams [13]. For FSP-LAPLACE, we use 100 context points placed on a regular grid and limit Lanczos to run at most 500 iterations. For FVI, we draw uniformly 100 context points at each update. For Laplace, we use the full generalized Gauss-Newton matrix and find the prior scale by maximizing the marginal likelihood [11].

**Ocean current modeling.** We consider the GulfDrifters dataset [31] and we follow the setup by Shalashilin [32]. We use as training data 20 simulated velocity measurements (red arrows in Figure 2), and consider as testing data 544 average velocity measurements (blue arrows in Figure 2) computed over a regular grid on the $[-90.8, -83.8] \times [24.0, 27.5]$ longitude-latitude interval. We standardize both features and regression targets. We incorporate physical properties of ocean currents into the models by applying the Helmholtz decomposition to the GP prior's covariance function [33] as well as to the neural network $f$ using the following parameterization

$$f(\cdot, \boldsymbol{w}) = \operatorname{grad} \Phi(\cdot, \boldsymbol{w}_1) + \operatorname{rot} \Psi(\cdot, \boldsymbol{w}_2) \tag{B.2}$$

where $\boldsymbol{w} = \{\boldsymbol{w}_1, \boldsymbol{w}_2\}$ and $\Phi(\cdot, \boldsymbol{w}_1)$ and $\Psi(\cdot, \boldsymbol{w}_2)$ are two hidden-layer fully-connected neural networks with hyperbolic tangent activation functions and 100 hidden units per layer. We use 96 context points placed on a regular grid for the FSP-LAPLACE and limit Lanczos to run at most 500 iterations. For FVI, we use the same 96 context points. For the linearized Laplace, we use the full generalized Gauss-Newton (GGN) and fix the prior scale to $\sigma_p = 1$.

**Image classification.** We consider the MNIST [34] and FashionMNIST [35] image classification datasets. We standardizing the images, fit the models on a random partition of $90\%$ of the provided train splits, keeping the remaining $10\%$ as validation data, and evaluate the models on the test data. We report mean and standard-errors across 5 such random partitions of the train data with different random seeds. We compare our model to FVI, Laplace, MAP, and Sparse GP baselines, as the scale of the datasets forbids exact GP inference. The expected log-likelihood and expected calibration error are estimated by Monte Carlo integration with 10 posterior samples. All neural networks have the same convolutional neural network architecture with three convolutional layers ($3 \times 3$ kernels and output channels 16, 32 and 64) interleaved with a max-pooling layer, before two fully connected layers (with output size 128 and 10). For FSP-LAPLACE, we sample context points from the Kuzushiji-MNIST (KMNIST) dataset [62] of $28 \times 28$ gray-scale images during training following Rudner et al. [26] and use $25'000$ points from the Halton low discrepancy sequence [63] to compute the covariance. We also consider sample context points uniformly over the range $[p_{min}^{h,w,c}, p_{max}^{h,w,c}]_{h=1,w=1,c=1}^{H,W,C}$, where $H$, $W$ and $C$ are respectively the height, width and number of channels of the images, and $p_{min}^{h,w,c} = v_{min}^{h,w,c} - 0.5 \times \Delta^{h,w,c}$ and $p_{max}^{h,w,c} = v_{max}^{h,w,c} + 0.5 \times \Delta^{h,w,c}$ where $\Delta^{h,w,c} = v_{max}^{h,w,c} - v_{min}^{h,w,c}$ is the difference between the minimal ($v_{min}^{h,w,c}$) and maximal ($v_{max}^{h,w,c}$) values of the data set at pixel index $(h, w, c)$. For FVI, we use the same context point distributions as FSP-LAPLACE. For the Laplace, we use the K-FAC approximation of the generalized Gauss-Newton matrix. We use a Categorical likelihood, the Adam optimizer [64] with a batch size of 100 and stop training early when the validation loss stops decreasing. We optimize hyper-parameters using the Bayesian optimization tool provided by Weights and Biases [65] and select the parameters which maximize the average validation expected log-likelihood across 1 random partitioning of the provided training split into training and validation data. We find covariance function parameters by maximizing the log-marginal likelihood from batches [66] using the reparameterization of classifications labels into regression targets from Milios et al. [61]. We optimize over kernel, prior scale, learning-rate, $\alpha_\epsilon$ (introduced by Milios et al. [61]) and activation function and select covariance functions among the RBF, Matern-1/2, Matern-3/2, Matern-5/2 and Rational Quadratic.

**Out-of-distribution detection with image data.** We consider out-of-distribution detection with image data and a Categorical likelihood following the setup by Osawa et al. [36]. We aim to partition in-distribution (ID) data from out-of-distribution (OOD) based on the mean of entropy of the predictive distribution with respect to 10 posterior samples using a single threshold found by fitting a decision stump. We evaluate a model fit on MNIST using its test data set as in-distribution data (ID) and the test data set of FashionMNIST as out-of-distribution (OOD), and vice-versa for a model fit on FashionMNIST. We use the same models and hyper-parameters as for image classification and report mean and standard-error of our scores across the same 5 random partitions of the data.

**Bayesian optimization.** We consider Bayesian optimization (BO) problems derived from Li et al. [7]. More specifically, we use the same setup but change the dimension of the feature space of the tasks. We report mean and standard error of 5 repetitions of the tasks across different random seeds. We use two hidden layer neural networks with hyperbolic tangent activations and 50 hidden units each. FSP-LAPLACE uses a Matern-5/2 covariance function with constant zero mean function whose parameters are found by maximizing the marginal likelihood [13]. We use 400 context points during

training and $10'000$ to compute the posterior covariance sampled using latin hypercube sampling [67]. We use the same prior for FVI and the same number of context points during training. The Gaussian process uses a zero mean function and a Matern-5/2 covariance function following Li et al. [7]. For the Laplace approximation, we find the prior scale by maximizing the marginal likelihood [11]. Unlike FSP-LAPLACE which uses low-rank factors to parameterize the posterior covariance, we found that repeatedly computing the covariance and predictive posterior of the linearized Laplace with the full and K-FAC generalized Gauss-Newton (GGN) matrices was often prohibitively slow in the BO setup. We therefore use the K-FAC approximation to the GGN where possible (Branin and PDE) and the diagonal approximation otherwise (Ackley, Hartmann, Polynomial and BNN). We implemented this experiment using the BO Torch library [68].

**Regression with UCI datasets.** We consider tabular regression datasets from the UCI repository [69]. Specifically, we perform leave-one-out 5-fold cross validation, considering $10\%$ of the training folds as validation data, and we report the mean and standard-error of the average expected log-likelihood on the test fold. We report the mean rank of the methods across all datasets by assigning rank 1 to the best scoring method as well as any method who's standard error overlaps with the highest score's error bars, and recursively apply this procedure to the methods not having yet been assigned a rank. We estimate the expected log-likelihood using 10 posterior samples. We encoding categorical features as one-hot vectors and standardizing the features and labels. We consider two hidden-layer neural networks with 50 hidden units each and hyperbolic tangent activations. All models have a homoskedastic noise model with a learned scale parameter. FSP-LAPLACE uses context points drawn uniformly over the range $[p_{min}^i, p_{max}^i]_{i=1}^d$, where $d$ is the dimension of the feature space, and $p_{min}^i = v_{min}^i - 0.5 \times \Delta^i$ and $p_{max}^i = v_{max}^i + 0.5 \times \Delta^i$ where $\Delta^i = v_{max}^i - v_{min}^i$ is the difference between the minimal ($v_{min}^i$) and maximal ($v_{max}^i$) values of the data set at feature index $i$. For the Laplace, we use the full generalized Gauss-Newton matrix. FVI uses the same context points as FSP-LAPLACE. Neural networks are fit using the Adam optimizer [64] and we stop training early when the validation loss stops decreasing. Hyper-parameters are found just as for the image classification experiment.

**Out-of-distribution detection with regression data.** We consider out-of-distribution (OOD) detection with tabular regression data from the UCI datasets [69] following the setup from Malinin et al. [70]. We aim to separate test data (in-distribution) from a subset of the song dataset [71] (out-of-distribution) with the same number of features based on the variance of the predictive posterior estimated from 10 posterior samples using a single threshold obtained via a decision stump. We process the data just like in the regression experiments, use the same model hyper-parameters and report mean and standard-error of the scores across the same 5 random partitions of the data.

### B.2.1 Software

We use the JAX [72] and DM-Haiku [73] Python libraries to implement neural networks. The generalized Gauss-Newton matrices used in the Laplace approximations are computed using the KFAC-JAX library [74]. We implemented the GPs and sparse GPs using the GPyTorch library [68]. We conducted experiments the Bayesian optimization experiments using the BOTorch library [68].

### B.2.2 Hardware

All models were fit using a single NVIDIA RTX 2080Ti GPU with 11GB of memory.

## C   Additional experimental results

### C.1   Additional qualitative results

**Regression.** We show additional results for the synthetic regression task described in Section 4. We find that FSP-LAPLACE successfully adapts to the beliefs specified by different Gaussian process priors in terms of periodicity (Figure C.2), smoothness (Figure 1) and length scale (Figure C.3) without modifying the neural network's architecture or adding features. We also find that our method effectively regularizes the model when the data is very noisy (Figure C.4).

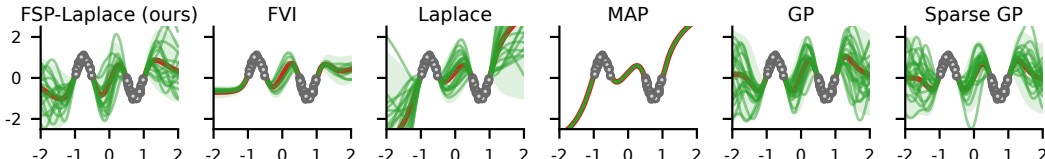

Figure C.1: Just like the Gaussian process (GP) and sparse GP, FSP-LAPLACE captures the smoothness behavior specified by the RBF covariance function of the Gaussian process prior.

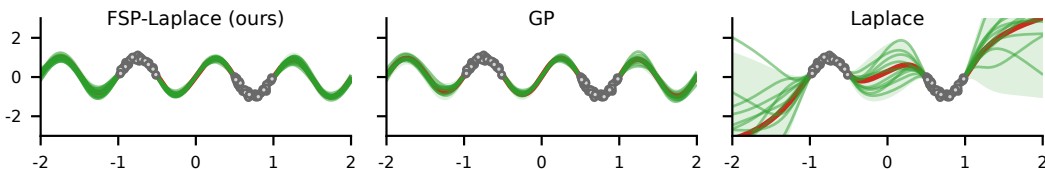

Figure C.2: Unlike the linearized Laplace, FSP-LAPLACE allows to incorporate periodicity within the support of the data using a periodic prior covariance function and without additional periodic features.

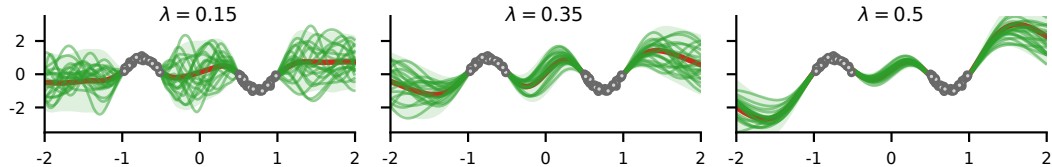

Figure C.3: FSP-LAPLACE adapts to the length scale provided by the RBF covariance function of the Gaussian process prior.

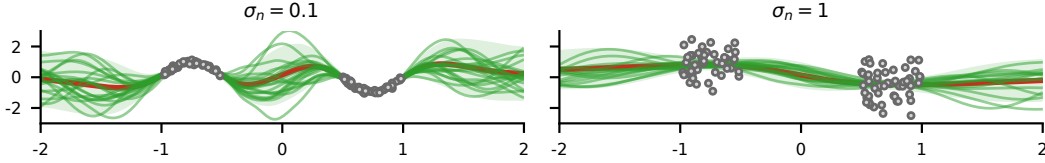

Figure C.4: FSP-LAPLACE is effectively regularized under strong label noise.

**Classification.** We show additional results for the two-moons classification task described in Section 4. Similar to the GP and sparse GP baselines, we find that our method captures the behavior of the prior, showing a smooth decision boundary when equipped with a RBF covariance function (Figure C.6) and a rough decision boundary when equipped with a Matern-1/2 covariance function (Figure C.5). FSP-LAPLACE also reverts to the zero-valued mean outside of the data support.

**Effect of context points.** We provide additional details on the role of the context points in FSP-LAPLACE. The goal of the context points is to regularize the neural network on a finite set of input locations which includes any point where we would like to evaluate the model. During MAP estimation (see Algorithm 1), context points are resampled at each update step to amortize the coverage of the feature space. During the posterior covariance computation (see Algorithm 2), context points are fixed and define where we regularize the model. Context points bare similarity with inducing points in variational GPs [27] in this later step as both define where the model is regularized.

We show additional results demonstrating the behavior of our model in the low context point regime. Figure C.7 shows the effect of the number of context points on a 1-dimensional regression task with GP priors equipped with a RBF and a Matern 1/2 kernel. Figure C.8 shows the same experiment but on a 2-dimensional classification task. The $M$ context points are randomly sampled uniformly during training, and we use the Halton low discrepancy sequence as context points to compute the covariance. Even with a very small number of context points ($M = 3$ and $M = 5$), our model still produces useful uncertainty estimates even if it cannot accurately capture the beliefs specified by the

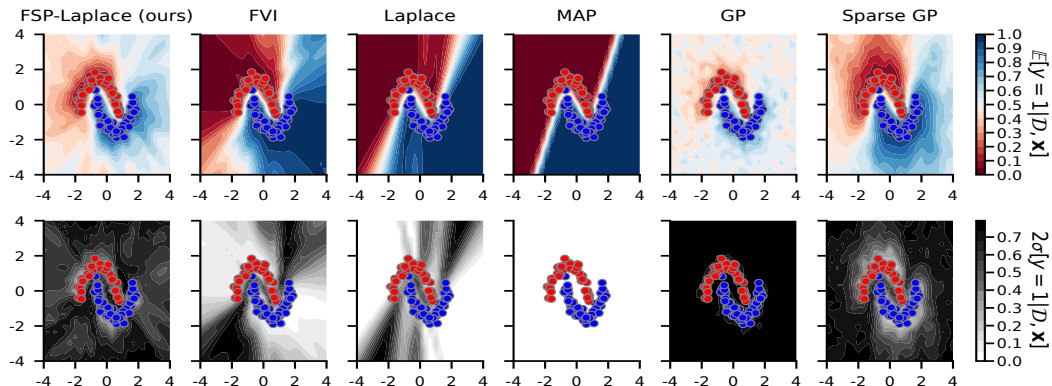

Figure C.5: FSP-LAPLACE with a Matern-1/2 covariance function against baselines in the two-moons classification task. Similar to the Gaussian process (GP) and sparse GP, our method shows a rough decision boundary.

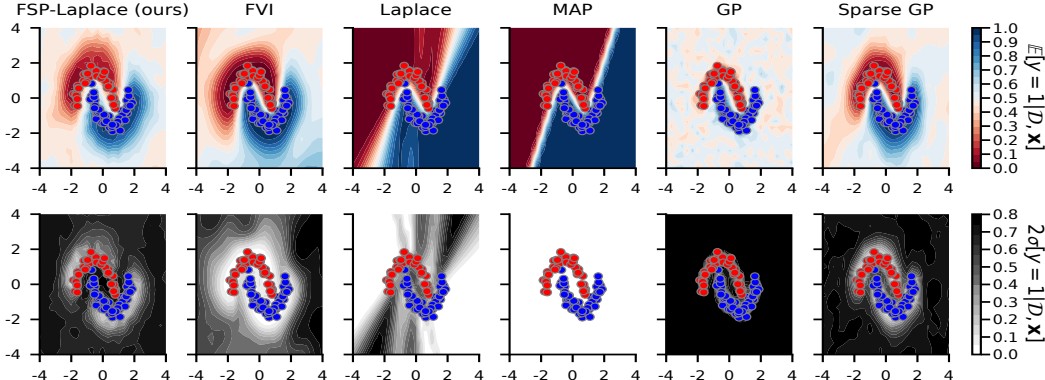

Figure C.6: FSP-LAPLACE with a RBF covariance function against baselines in the two-moons classification task. Similar to the Gaussian process (GP) and sparse GP, our method shows a smooth decision boundary.

prior. We also note that our method requires more context points to capture the beliefs of rougher priors than smooth priors (see Figure C.7).

## C.2 Additional quantitative results

**Mauna Loa.** We here show the figure associated with the Mauna Loa dataset experiment in Section 4.1. Figure C.9 shows the predictions of FSP-LAPLACE and the baselines on the Mauna Loa dataset. We find that incorporating prior beliefs both via an informative prior and periodic features in FSP-LAPLACE results in an improved fit over FVI, Laplace and GP baselines.

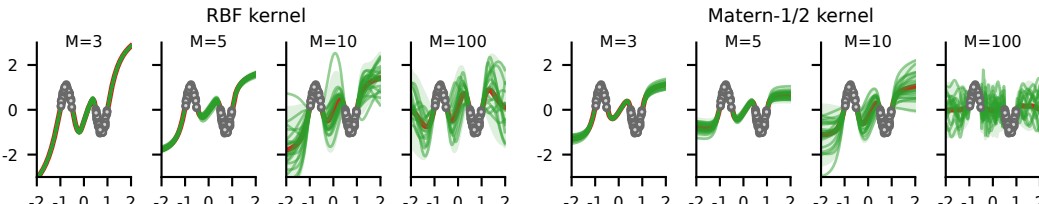

Figure C.7: FSP-LAPLACE with a smooth RBF covariance function and a rough Matern-1/2 with varying amounts of context points $M$. Given a very small number of context points, our method still produces useful uncertainty estimates.

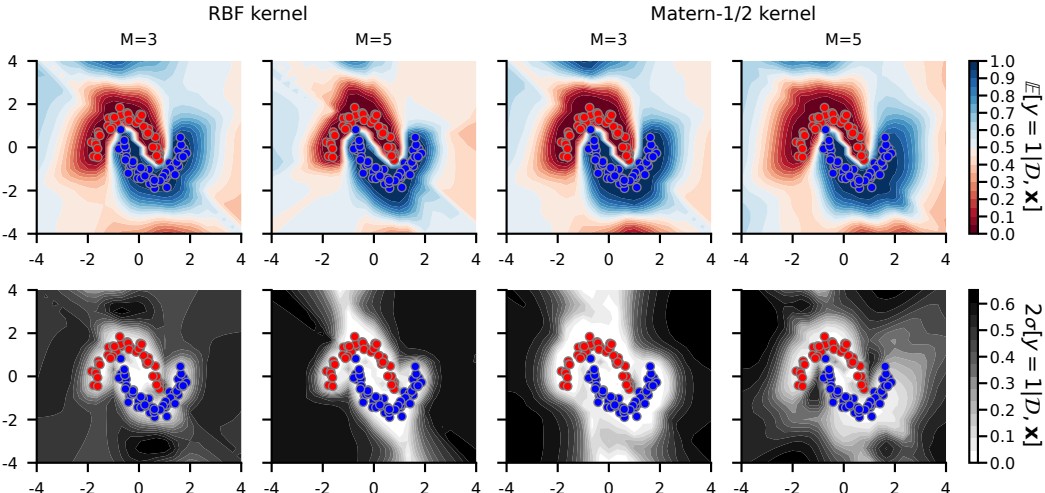

Figure C.8: FSP-LAPLACE with a smooth RBF covariance function and a rough Matern-1/2 with varying amounts of context points $M$. Given a very small number of context points, our method still produces useful uncertainty estimates and reverts to the prior's zero valued mean.

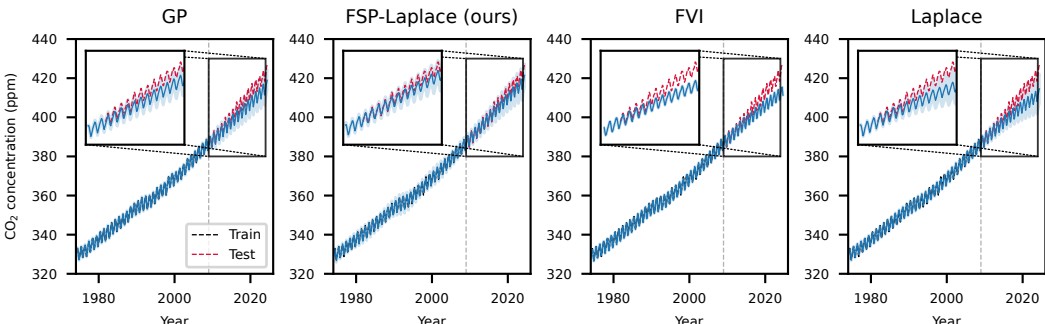

Figure C.9: Regression on the Mauna Loa dataset. Incorporating prior knowledge via a kernel tailored specifically to the dataset, our method (FSP-LAPLACE) results in a strong decrease in mean square error over baselines.

**Out-of-distribution detection with image data.** We present additional results for the out-of-distribution detection experiment presented in Section 4.1. Figure C.10 shows the distribution of the predictive entropy of in-distribution (ID) and out-of-distribution (OOD) data under the sparse GP, FSP-LAPLACE, FVI and Laplace models. For both MNIST and FashionMNIST, we find that the predictive entropy of ID data under FSP-LAPLACE is tightly peaked around $0$ nats and that the predictive entropy of OOD data strongly concentrates around its maximum $\ln 10 \approx 2.3$ nats.

**Rotated MNIST and FashionMNIST.** We provide an additional experiment studying the behavior of FSP-LAPLACE under out-of-distribution data. We consider the setup by Rudner et al. [25], Sensoy et al. [75] and track the predictive entropy of models trained on MNIST and FashionMNIST for increasing angles of rotation of the test images. We expect the predictive entropy of a well-calibrated neural network to grow as the inputs become increasingly dissimilar to the training data with higher angles of rotation. Similar to FVI, sparse GP and the linearized Laplace baselines, we find that FSP-LAPLACE yields low predictive entropy for small rotation angles and that the predictive entropy increases with the angle on both MNIST and FashionMNIST which is what we expect from a well calibrated Bayesian model.

**Regression with UCI datasets.** We further evaluate our method on regression datasets from the UCI repository [69] and compare FSP-LAPLACE to FVI, Laplace, MAP, GP, and Sparse GP baselines. We perform leave-one-out 5-fold cross-validation, keeping $20\%$ of the remaining train folds as validation data. We report the mean and standard-error of the expected log-likelihood with respect to samples

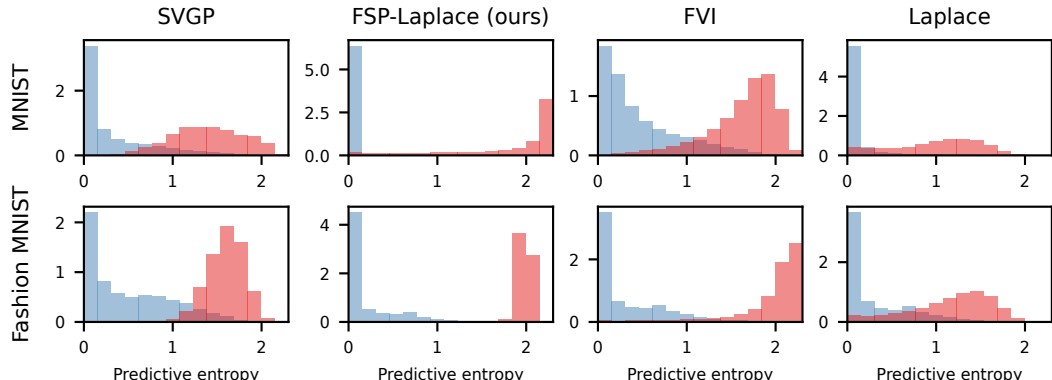

Figure C.10: Distribution of in-distribution (ID) and out-of-distribution (OOD) samples for the MNIST and Fashion MNIST image datasets. The predictive entropy produced by FSP-LAPLACE nearly perfectly partition ID and OOD data. This is reflected in OOD accuracy in Table 2.

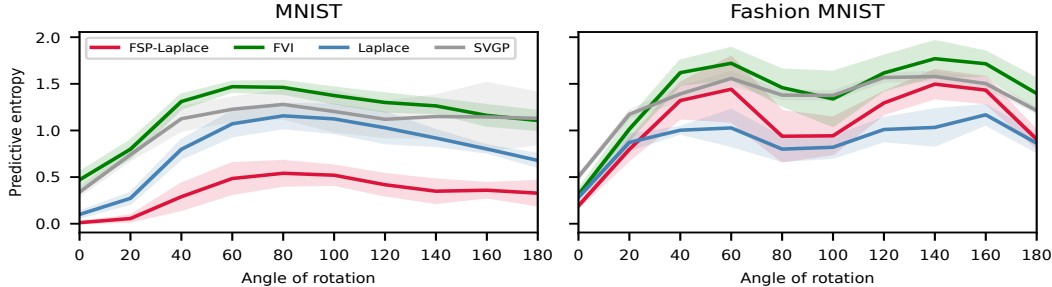

Figure C.11: Input distribution shift experiment. We find that FSP-LAPLACE shows greater predictive entropy as the input becomes increasingly dissimilar to the training data.

from the posterior across the 5-folds. Additional details can be found in Appendix B.2. Results are presented in Table C.1. We find that our method performs well compared to baselines, matching or improving over the mean rank of all Bayesian methods (FVI, Laplace, GP, and Sparse GP) but a slightly lower mean rank than the MAP baseline (1.636 vs. 1.363). In particular, our method is noticeably more accurate than the Laplace.

Table C.1: Test log-likelihood and out-of-distribution accuracy of evaluated methods on regression datasets from the UCI repository. We find that our method matches or improves over the mean rank of Bayesian baselines in terms of expected log-likelihood, and obtains out-of-distribution detection accuracies similar to the best performing BNN.

| DATASET | EXPECTED LOG-LIKELIHOOD (↑) | | | | | | OUT-OF-DISTRIBUTION DETECTION ACCURACY (↑) | | | | |
| --- | --- | --- | --- | --- | --- | --- | --- | --- | --- | --- | --- |
| | FSP-LAPLACE (OURS) | FVI | LAPLACE | MAP | GP | SPARSE GP | FSP-LAPLACE (OURS) | FVI | LAPLACE | GP | SPARSE GP |
| BOSTON | **-0.641 ± 0.084** | **-0.512 ± 0.105** | -0.733 ± 0.019 | -0.906 ± 0.274 | -1.136 ± 0.035 | -0.762 ± 0.077 | 0.923 ± 0.015 | 0.754 ± 0.019 | 0.913 ± 0.008 | 0.948 ± 0.004 | **0.969 ± 0.003** |
| CONCRETE | -0.352 ± 0.052 | -0.378 ± 0.076 | -0.538 ± 0.026 | **-0.199 ± 0.056** | -0.766 ± 0.119 | -0.609 ± 0.050 | **0.914 ± 0.007** | 0.585 ± 0.011 | 0.859 ± 0.002 | 0.878 ± 0.005 | 0.854 ± 0.007 |
| ENERGY | 1.403 ± 0.033 | 1.448 ± 0.059 | 1.423 ± 0.062 | **1.754 ± 0.058** | 1.370 ± 0.060 | 1.518 ± 0.032 | **1.000 ± 0.000** | 0.928 ± 0.007 | **1.000 ± 0.000** | **1.000 ± 0.000** | **1.000 ± 0.000** |
| KIN8NM | -0.134 ± 0.010 | **-0.135 ± 0.021** | -0.199 ± 0.009 | **-0.105 ± 0.011** | - | -0.206 ± 0.003 | 0.626 ± 0.006 | 0.593 ± 0.006 | 0.609 ± 0.015 | - | **0.645 ± 0.004** |
| NAVAL | **3.482 ± 0.014** | 2.943 ± 0.062 | 3.213 ± 0.028 | **3.460 ± 0.048** | - | **3.474 ± 0.019** | **1.000 ± 0.000** | **1.000 ± 0.000** | **1.000 ± 0.000** | - | **1.000 ± 0.000** |
| POWER | 0.045 ± 0.014 | **0.071 ± 0.018** | 0.016 ± 0.011 | **0.047 ± 0.013** | - | 0.026 ± 0.009 | **0.818 ± 0.003** | 0.686 ± 0.014 | **0.820 ± 0.003** | - | 0.777 ± 0.004 |
| PROTEIN | **-0.991 ± 0.005** | **-0.988 ± 0.002** | -1.043 ± 0.006 | -1.003 ± 0.003 | - | -1.035 ± 0.002 | 0.917 ± 0.004 | 0.857 ± 0.015 | **0.992 ± 0.001** | - | 0.967 ± 0.001 |
| WINE | **-1.189 ± 0.019** | **-1.193 ± 0.015** | -1.403 ± 0.014 | **-1.174 ± 0.022** | -1.313 ± 0.032 | -1.194 ± 0.020 | 0.591 ± 0.024 | 0.541 ± 0.012 | 0.763 ± 0.012 | **0.812 ± 0.008** | 0.680 ± 0.014 |
| YACHT | 0.183 ± 0.438 | 0.655 ± 0.293 | 0.136 ± 0.762 | **1.644 ± 0.310** | 1.336 ± 0.060 | 1.532 ± 0.139 | **0.973 ± 0.008** | 0.594 ± 0.010 | 0.851 ± 0.029 | 0.842 ± 0.023 | 0.651 ± 0.011 |
| WAVE | 6.112 ± 0.049 | 6.552 ± 0.155 | 6.460 ± 0.025 | **7.146 ± 0.195** | - | 4.909 ± 0.001 | **0.912 ± 0.076** | **0.861 ± 0.014** | 0.810 ± 0.023 | - | 0.513 ± 0.001 |
| DENMARK | **-0.364 ± 0.008** | -0.462 ± 0.006 | -0.810 ± 0.010 | -0.693 ± 0.014 | - | -0.768 ± 0.001 | 0.677 ± 0.004 | 0.634 ± 0.008 | **0.714 ± 0.006** | - | 0.626 ± 0.002 |
| MEAN RANK | 1.636 | 1.636 | 2.727 | 1.363 | 2.600 | 2.455 | 1.818 | 3.091 | 1.727 | 1.6 | 2.182 |

**Out-of-distribution detection on tabular data.** We also investigate whether the epistemic uncertainty of FSP-LAPLACE is predictive of out-of-distribution data in the regression setting by evaluating it on out-of-distribution detection following Malinin et al. [70]. We report the accuracy of a single threshold to classify OOD from in-distribution (ID) data based on the predictive uncertainty. More details are presented in Appendix B.2. In the context of tabular data, we find that FSP-LAPLACE performs second best among BNNs and is almost as accurate as the Laplace approximation (see Table C.1) which is first. We note that FSP-LAPLACE systematically outperforms FVI in terms of out-of-distribution detection and obtains a higher mean rank (1.818 vs. 3.091).

$\|P_0 \Lambda P_0^\top\|_F$ **is negligible.** We provide evidence that the term $\|P_0 \Lambda P_0^\top\|_F$ in Section 3.2 is negligible compared to $\|\Lambda\|_F$ in four different configurations. We consider the synthetic regression and classification setups described in Appendix B.1.

Table C.2: $\|P_0 \Lambda P_0^\top\|_F / \|\Lambda\|_F$ for different combinations of priors and tasks.

| KERNEL | REGRESSION | CLASSIFICATION |
|---|---|---|
| RBF | $1.026 \times 10^{-7}$ | $3.548 \times 10^{-4}$ |
| MATERN-1/2 | $4.305 \times 10^{-6}$ | $1.299 \times 10^{-2}$ |

