# OpenReview forum: "FSP-Laplace: Function-Space Priors for the Laplace Approximation in Bayesian Deep Learning"
_NeurIPS.cc/2024/Conference — NeurIPS 2024 poster_

### Official Review · Reviewer_KV8g · 2024-07-12

**Soundness:** 1
**Presentation:** 3
**Contribution:** 2
**Rating:** 2
**Confidence:** 4

**Summary:**

This work takes a function space view of the posterior distribution, and places a GP on the likelihood with neural network predictors. This generate a posterior distribution, and the authors attempt a Laplace approximation as a which they combine with matrix-free linear algebraic methods to aid in computational tractability. Numerical examples are provided demonstrating their method performs well.

**Strengths:**

This paper is considers an important problem -- high quality uncertainty estimates for neural models are an important area of machine learning research, and crucial when deploying models in critical systems.
The work is generally well written and presented.
The method performs well in the numerics section.

**Weaknesses:**

Laplace approximations are justified by the Bernstein-von Mises theorem, which states that (under appropriate assumptions) the posterior distribution concentrates around its mode \emph{independent of the prior distribution} in the large-data limit.
In order to use a Laplace approximation out of the box, it's important for these assumptions to be met, and the model to be in the appropriate pre-limit.
Singular learning theory is one approach that addresses this for singular models such as neural networks (see https://www.routledge.com/Mathematical-Theory-of-Bayesian-Statistics/Watanabe/p/book/9780367734817).
However approximating the Hessian, or regularizing it and claiming it's because there's a prior not only does nothing to fix the problem, but perpetuates a line of ill-posed research.
While taking a function space view opens up the possibility of priors that concentrate with the dataset size and therefore cannot be discarded in the BvM limit, the authors claim that the eigenvalues of their prior term rapidly decay and discard them, killing any possibility that they have considered this concentration.
other forms of Laplace approximations may be more suitable (see https://arxiv.org/pdf/2110.12922, https://arxiv.org/abs/2307.07785 for related methods using expansions in data-space).
Having said this, the work does produce decent empirical results as a quadratic approximation of the posterior \emph{inspired by} Laplace approximations, however the current reasoning used to get there is ill-considered.

The dependence of $f$ appearing in Propositions 1 and 2 should be made explicit.

The proofs given in the paper are not sound. In particular, the proof of Proposition 2 is "Analogous to the proof of Theorem 5.2(b) from Lambley [20].". However, Theorem 5.2(b) in [20] is not proved, where it is stated that the proof is similar to another reference. That reference also points to the proof of another corollary, stating it is analogous. A chain this long suggests not only that the proof needs to be written out in the current context to concretely check that the 3 analogies used are sound, but that the authors have not bothered to check the references themselves, let alone write out the proofs internally. The proof of Proposition 1 is also not clear, and needs to be written out, so that it is not burdensome for the reader to confirm the claims.

The claim that the term appearing in line 195 is negligible should be confirmed numerically.

**Questions:**

Can you please address the weaknesses above?

**Limitations:**

No, limitations were not adequately addressed. However, there is little view for negative societal impact in work of this nature.

---

> ### Author Rebuttal · Authors · 2024-08-07
>
> We thank the reviewer for their time and feedback. We are happy that they found the paper was "well written" and that our method "performs well".
>
> > "Laplace approximations are justified by the Bernstein-von Mises theorem  which states that [...] the posterior distribution concentrates around its mode *independent of the prior distribution* in the large-data limit. [...] However approximating the Hessian, or regularizing it and claiming it's because there's a prior not only does nothing to fix the problem, but perpetuates a line of ill-posed research."
>
> We respectfully disagree. It is true that the Bernstein-von Mises theorem only holds in the large-data limit. However, there is a long line of empirical work suggesting that the linearized Laplace approximation works surprisingly well for neural networks despite the lack of theoretical justification [(Papamarkou et al. 2024, Section 3.1 references many such papers)](https://proceedings.mlr.press/v235/papamarkou24b.html). Our paper provides additional empirical evidence supporting this observation. Calling this entire line of research "ill-posed" needs to be measured against the large amount of empirical evidence that clearly demonstrates the value of such research.
>
> More generally, the point of empirical research is precisely to explore regimes in which thoeretical statements are not possible. Rigorous theoretical statements can often only be proven under strong assumptions, but this does not mean that the opposite of the statement is true as soon as assumptions are violated (absence of proof is not proof of absence). Empirical research is indispensable to explore how far away from a perfect limiting case remnants of a theory still apply. For example, the entire field of deep learning relies on optimization methods for which most of the theory applies only to convex functions, and yet methods motivated by this theory are successfully used to train highly non-convex deep neural networks because empirical studys show that they work surprisingly well in certain useful regimes far beyond convexity. This is not an argument against the value of theory. On the contrary: it would have been impossible for us to "guess" the empirically successful approximation scheme that we propose had we not been guided by theory that strictly holds only in a limiting case that is admittedly relatively far away from our applications.
>
> > "the work does produce decent empirical results as a quadratic approximation of the posterior *inspired by* Laplace approximations, however the current reasoning used to get there is ill-considered."
>
> We thank the reviewer for acknowledging our empirical results. We agree that these results were obtained with a method that is inspired by Laplace approximations (see our proofs of Propositions 1 and 2 in the additional comment for clarification on this connection). As with most forms of scalable Bayesian inference, several additional approximations were necessary to make our method usable in practice. We believe that our paper clearly highlights the additional approximations, e.g., by explicitly grouping them into Section 3.2. These additional approximations do not change the fact that the resulting algorithm approximates the posterior with a Gaussian distribution whose mean is the MAP (of the RKHS-regularized neural network) and whose precision matrix is obtained by approximating the curvature at this point. Methods for approximate posterior inference that follow this scheme are generally called Laplace approximations in the literature.
>
> > "The dependence of f appearing in Propositions 1 and 2 should be made explicit."
>
> We do not understand your inquiry, would you mind elaborating on what you mean by the "dependence on f appearing in Propositions 1 and 2?
> From our point of view, there is no dependence on any particular $\mathbf{f}$ in Propositions 1 and 2, since in these propositions $\mathbf{f}$ is an unbound variable denoting an arbitrary element of the sample space $\mathbb{B}$ of the GP prior (or an element of the RKHS $H_{\mathbf{\Sigma}} \subset B$ depending on the context).
> Moreover, the $\mathbf{f}^\star_n$ in Proposition 2 are explicitly defined as minimizers of a sequence of optimization problems, and $\mathbf{f}^\star$ is also explicitly defined as the weak limit of a certain subsequence of $( \mathbf{f}^\star_n )_{n \in N}$.
>
> > "The proofs given in the paper are not sound. In particular, the proof of Proposition 2 [...]. The proof of Proposition 1 is also not clear, and needs to be written out"
>
> We agree that the proof of Propositions 2 is difficult to follow as one has to combine several ideas in the referenced works in a nontrivial manner.
> Hence, we conducted the proof of Proposition 2 in full detail and attach it as a separate comment below.
> It will also be included in the appendix in the camera-ready version of the paper.
>
> Under Assumptions A.1 to A.3 detailed in the paper, Proposition 1 is a pretty direct corollary of Theorem 1.1 in (Lambley, 2023).
> The given proof in the paper verifies the assumptions of Lambley's Theorem 1.1 and is hence complete.
> However, we will make an effort in the camera-ready version of the paper to add more explanatory comments to make the proof easier to follow.
>
> > "The claim that the term appearing in line 195 is negligible should be confirmed numerically."
>
> We provide evidence that the term $P_0 \Lambda P_0^\top$ is negligible in four different configurations.
> First, considering the regression setup described in Appendix B.1, we have:
> * RBF kernel: $\frac{||P_0 \Lambda P_0^\top||_F}{||\Lambda||_F} = 1.026 \times 10^{-7}$
> * Matern-1/2: $\frac{||P_0 \Lambda P_0^\top||_F}{||\Lambda||_F} = 4.305 \times 10^{-6}$
>
> Considering the classification setup described in Appendix B.1, we have:
> * RBF kernel: $\frac{||P_0 \Lambda P_0^\top||_F}{||\Lambda||_F} = 3.548 \times 10^{-4}$
> * Matern-1/2: $\frac{||P_0 \Lambda P_0^\top||_F}{||\Lambda||_F} = 1.299 \times 10^{-2}$

---

> > ### Comment · Reviewer_KV8g · 2024-08-12
> > **Response**
> >
> > Thanks for your response, however it is still unclear if you intend to address any of these concerns in the document.
> >
> > I think that the analogy between the construction of so-called "Laplace approximations" and the failure of optimization techniques to guarantee convergence outside convex settings is incorrect. First, one can obtain results under relaxations of convexity. However, I believe that the authors were referring to guarantees of convergence to local vs global minima. This is not the same as the situation in the Laplace approximation case, where the construction is known to be incorrect, and a number of alternative constructions have been provided in relevant settings that should not be ignored. The authors do acknowledge that there are issues with the construction. Addressing these limitations up front are in the best interest of both the author, and the field at large. The alternative seems to be acknowledging the flaws in private but trying to sweep things under the rug for the purposes of publication. This is a particular issue in the LA setting, where the empirical results obtained are usually poor when compared to ensembling, which is presumably why the methods are rarely compared to network ensembles. The authors have also not addressed my concerns regarding the decaying eigenvalues of the prior term.
> >
> > My complaint about the proof of Proposition 2 was not that it was difficult to follow, merely that it did not appear. It is reasonably straightforward linear analysis, but the burden should not be placed upon the reader to chase through multiple resources to find something resembling a proof of a stated proposition . The authors finding it difficult to follow is not good justification for its omission. I am glad the mathematics will appear in the updated version, and Proposition 1 will be clarified.
> >
> > The numerics regarding $P_0 \Lambda P_0^\top$ are reassuring. I hope that the authors intend to include them in their updated work, however no indication has been given that this is the case.
> >
> > I would like to state that this is a field largely driven by empirical results. However, the framing of this document is as a theoretically driven and quite technical construction, which as I have outlined is not sound. This would not be an issue, except that the authors seem unwilling to change the tone and framing of the work to address its limitations. Since there has been no indication that they intend to address these issues, I cannot change my score at this stage.

---

> > > ### Author Response · Authors · 2024-08-14
> > >
> > > Thank you for your response and clarifications.
> > >
> > > > [comparison to deep ensembles]
> > >
> > > Thank you for bringing up deep ensembles.
> > > Unfortunately, this concrete criticism comes up too late in the review process for us to run additional experiments a this point.
> > >
> > > We first wish to highlight that our contribution is a method to specify informative prior beliefs on the function represented by the neural network in the framework of the Laplace approximation.
> > > Deep ensembles typically use isotropic Gaussian priors on the weights (i.e. $l_2$ regularization) and we are unaware of any method to pose informative function space priors in neural network ensembles.
> > > While interesting, we believe that comparing our method to deep ensembles is not directly relevant for our paper as they are neither related to the Laplace approximation nor to function-space priors.
> > > Just like the other baselines, we expect our method to outperform deep ensembles in cases where a GP accurately reflects prior beliefs.
> > >
> > > Outside of the discussion on priors, we further wish to highlight that deep ensembles are also much more expensive to fit than Laplace approximations.
> > > Training multiple neural networks entirely from scratch is unthinkable in large neural networks and in applications where the models are often updated (e.g., Bayesian optimization).
> > > The diversity of predictive ensembles is also known to collapse when the size of the neural network increases ([Abe et al., 2024](https://arxiv.org/abs/2302.00704)).
> > >
> > > Finally, in contrast to deep ensembles, Laplace approximations provide a parametric (approximate) posterior probability distribution over the weights of the BNN, which is useful for more than just predictive uncertainty quantification (e.g., weight pruning, model compression, continual learning).
> > >
> > > > I would like to state that this is a field largely driven by empirical results. However, the framing of this document is as a theoretically driven and quite technical construction, which as I have outlined is not sound. This would not be an issue, except that the authors seem unwilling to change the tone and framing of the work to address its limitations.
> > >
> > > Thank you for acknowledging that the field is "largely driven by empirical results".
> > > We believe that our results provide strong empirical evidence that our method effectively incorporates beliefs specified by a GP prior and provides sensible uncertainty estimates across many applications.
> > >
> > > We respectfully disagree that the paper is "theoretically driven".
> > > While we include theory to motivate and justify our method, our contribution is a practical algorithm.
> > >
> > > Nevertheless, we are willing to take your criticism and we plan to allocate a part of the additional page allowed for the camera ready version to provide more intuition about the method, and we will include a brief summary of empirical results early on in the paper (pointing to the extended results section for details).
> > >
> > > > The authors have also not addressed my concerns regarding the decaying eigenvalues of the prior term.
> > > > [Comment referred to above: While taking a function space view opens up the possibility of priors that concentrate with the dataset size and therefore cannot be discarded in the BvM limit, the authors claim that the eigenvalues of their prior term rapidly decay and discard them, killing any possibility that they have considered this concentration.]
> > >
> > > As stated in our original rebuttal, we do not consider the BvM theorem to be the justification of our method.
> > > Hence, we do not consider the decaying eigenvalues of the covariance matrices of the finite-dimensional marginals of the prior to be concerning.
> > >
> > > > My complaint about the proof of Proposition 2 was not that it was difficult to follow, merely that it did not appear. It is reasonably straightforward linear analysis, but the burden should not be placed upon the reader to chase through multiple resources to find something resembling a proof of a stated proposition . The authors finding it difficult to follow is not good justification for its omission. I am glad the mathematics will appear in the updated version, and Proposition 1 will be clarified.
> > >
> > > We are frankly having difficulties understanding this paragraph. In the original review, the reviewer asked for a proof of Proposition 2. We provided this proof in the comment above and promised to include it in the appendix of the camera ready version. We take the reviewer's statement that the proof is "reasonably straightforward linear analysis" as a confirmation that they did not find an error in our proof. We would have expected that this resolves the issue.
> > >
> > > > The numerics regarding $P_0 \Lambda P_0^\top$ are reassuring. I hope that the authors intend to include them in their updated work, however no indication has been given that this is the case.
> > >
> > > We will naturally include the numerical results supporting that $P_0 \Lambda P_0^\top$ is negligible in the Appendix of the camera ready version of the paper.

---

> ### Author Response · Authors · 2024-08-07
> **Proof of Proposition 2**
>
> The markdown interpreter does not seem to work correctly in math mode. We apologise for this and can provide a PDF with the proof upon request.
>
> **Proposition 2.**
> *Let Assumptions A.1 to A.3 hold.*
>
> *Let $\{\lambda_n\}_{n \in N} \subset R_{> 0}$ with $\lambda_n \to 0$, and $\{ f^\star_n \}_{n \in \N} \subset H_{\Sigma}$ such that $f^\star_n$ is a minimizer of $\R_\text{FSP}^{\lambda_n}$.*
>
> Then
>
> $\{f_n^\star\}_{n \in \mathbb{N}}$
>
> has an $H_{\Sigma}$-weakly convergent subsequence with limit
>
> $f^\star \in F$.
>
> Our proof makes use of ideas from (Dashti et al., 2013) and (Lambley, 2023).
>
> *Proof.*
> Without loss of generality, we assume $\mathbf{\mu} = \mathbf{0}$.
> By Assumption A.2, there are constants $K, \alpha > 0$ such that
>
> $R_\text{FSP}(\mathbf{f}) \ge K + \alpha \lVert \mathbf{f} \rVert_{H_{\mathbf{\Sigma}}}^2$
>
> for all $\mathbf{f} \in H_{\mathbf{\Sigma}}$ (Lambley, 2023, Section 4.1).
> Now fix an arbitrary $\mathbf{f} \in H_{\mathbf{\Sigma}} \cap F$.
> Then
>
> $R_\text{FSP}(\mathbf{f}) = R_\text{FSP}^{\lambda_n}(\mathbf{f})$
>
>
> $\ge R_\text{FSP}^{\lambda_n}(\mathbf{f}^\star_n)$
>
>
> $= R_\text{FSP}(\mathbf{f}^\star_n) + \lambda_n^{-1} d_{B}(\mathbf{f}^\star_n, F)$
>
>
> $\ge K + \alpha \lVert \mathbf{f}^\star_n \rVert_{H_{\mathbf{\Sigma}}}^2 + \lambda_n^{-1} d_{B}(\mathbf{f}^\star_n, F)$
>
>
> $\ge K + \alpha \lVert \mathbf{f}^\star_n \rVert_{H_{\mathbf{\Sigma}}}^2,$
>
> and hence
>
> $\lVert \mathbf{f}^\star_n \rVert_{H_{\mathbf{\Sigma}}}^2
> \le \frac{1}{\alpha} \left( R_\text{FSP}(\mathbf{f}) - K \right),$
>
> i.e. the sequence $\{f^\star_n\}_{n \in \mathbb{N}} \subset H_{\Sigma}$ is bounded.
>
> By the Banach-Alaoglu theorem (Conway, 1997, Theorems V.3.1 and V.4.2(d)) and the Eberlein-Šmulian theorem (Conway, 1997, Theorem V.13.1), there is a weakly convergent subsequence $\{ \mathbf{f}^\star_{n_k} \}_{k \in \N}$ with limit $\mathbf{f}^\star \in H_{\mathbf{\Sigma}}$.
>
> It remains to show that $\mathbf{f}^\star \in F$.
> From the inequality above, it follows that
>
> $0 \le d_{B}(\mathbf{f}^\star_{n_k}, F) \le \lambda_{n_k} \left( R_\text{FSP}(\mathbf{f}) - K - \alpha \lVert \mathbf{f}^\star_n \rVert_{H_{\mathbf{\Sigma}}}^2 \right),$
>
> where the right-hand side converges to 0 as $k \to \infty$.
>
> Hence, $\lim_{k \to \infty} d_{B}(\mathbf{f}^\star_{n_k}, F) = 0$.
>
> The embedding $\iota \colon H_{\mathbf{\Sigma}} \to B$ is compact (Bogachev, 1998, Corollary 3.2.4) and hence $\{\iota[\mathbf{f}^\star_{n_k}] \}_{k \in \mathbb{N}} \subset B$ converges ($B$-strongly) to $\iota[f^\star] \in B$ (Conway, 1997, Proposition 3.3(a)).
>
> The continuity of $d_{B}(\,\cdot\,, F)$ implies
>
> $d_{B}(\mathbf{f}^\star, F)= d_{B}(\iota[\mathbf{f}^\star], F)$
>
>
> $= d_{B} \left( \lim_{k \to \infty} \iota[\mathbf{f}^\star_{n_k}], F \right)$
>
>
> $= \lim_{k \to \infty} d_{B}(\iota[\mathbf{f}^\star_{n_k}], F)$
>
>
> $= \lim_{k \to \infty} d_{B}(\mathbf{f}^\star_{n_k}, F)$
>
>
> $= 0,$
>
> and by Assumption A.3 and Lemma A.2 it follows that $\mathbf{f}^\star \in F$.

---

### Official Review · Reviewer_ospn · 2024-07-12

**Soundness:** 3
**Presentation:** 3
**Contribution:** 3
**Rating:** 8
**Confidence:** 4

**Summary:**

This paper proposes a functional Laplace approximation approach which is able to incorporate function space prior instead of weight space prior used by existing Laplace approximation methods. The function space prior is more meaningful than weight space ones in terms of expressing prior knowledge about the underlying functions. The core idea of this paper is, on top of [20], to resolve the problem of functional probability density issue of naïve extension to functional MAP. The results comparing to weight space LA show the effectiveness of the new functional LA.

**Strengths:**

The paper is well-written and it is easy to follow the idea. There are sufficient details and background information to understand each part.
The method is reasonably designed and the flow is smooth.

**Weaknesses:**

The comparative experiments are not sufficient. Various efficient LA methods should be compared.

Ortega, L.A., Santana, S.R. and Hernández-Lobato, D., 2023. Variational linearized Laplace approximation for Bayesian deep learning. arXiv preprint arXiv:2302.12565.

McInerney, J. and Kallus, N., 2024. Hessian-Free Laplace in Bayesian Deep Learning. arXiv preprint arXiv:2403.10671.

Deng, Z., Zhou, F. and Zhu, J., 2022. Accelerated linearized Laplace approximation for Bayesian deep learning. Advances in Neural Information Processing Systems, 35, pp.2695-2708.

It would be better to include some functional BNN as well.

There are some typos, like Line89 and Line94.

**Questions:**

Can you please explain the equation in Line148 in more details?

What is the meaning of 'rigorous probabilistic interpretation' in Line 128? Does that mean the new P_B(df) is a properly defined functional posterior?

The objective function is similar to the kernel ridge regression. Can you please explain the difference?

Can you please add comparative results with functional BNN?

---

> ### Author Rebuttal · Authors · 2024-08-07
>
> We thank the reviewer for their time and positive feedback. We are glad that they found our paper was "well-written", "easy to follow" and had a "smooth flow".
>
> > "The comparative experiments are not sufficient. Various efficient LA methods should be compared. [...] It would be better to include some functional BNN as well."
> > "Can you please add comparative results with functional BNN?"
>
> We agree with the reviewer that a comparison with a functional BNN would be useful. Therefore, we ran our experiments on FVI (Sun et al., 2019) and present the results in the general rebuttal (see Figures D.1 and D.2 and Tables D.1, D.2 and D.3). We find that our method generally outperforms FVI.
>
> We use the full Laplace posterior covariance for regression, Mauna Loa, and ocean current modeling; due to computational limitations, we use only the KFAC approximation for the classification experiments and for Bayesian optimization. KFAC is standard in Laplace approximations for neural network (Ritter et al. 2018, Immer et al. 2021) and has shown to perform very competitively. The methods proposed by the reviewer offer more scalable alternatives when the full GGN is not available. While these are interesting in themselves, the point of this paper is to discuss the influence of a function-space prior and not of the GGN approximation. Exploring the use of the methods proposed by the reviewer to make our method more scalable sounds like an interesting direction for future work.
>
> > "Can you please explain the equation in Line148 in more details?"
>
> The equation follows from inserting the definition of $f^\text{lin}$ (Eq. 2.2) into the left-hand side and completing the square. We note that the $\dagger$ symbol designates the pseudo-inverse.
> In detail:
>
> <!--$$\frac{1}{2} ||f^{\text{lin}}(\,\cdot\ , \mathbf{w}) - \mathbf{\mu}||_{\mathbb{\mathbf{\Sigma}}}^2 = \frac{1}{2} (\mathbf{w} - \mathbf{w}^*)^\top \mathbb{\mathbf{\Sigma}}_{\mathbf{w}^*}^\dagger(\mathbf{w} - \mathbf{w}^*) + \mathbf{v}\mathbf{v}^\top (\mathbf{w} - \mathbf{w}^*) + \frac{1}{2} ||f(\,\cdot\ ,  \mathbf{w}^*) - \mathbf{\mu}||_\mathbb{\mathbf{\Sigma}}^2$$ -->
> from which we complete the square to obtain the right-hand side of line 148.
>
> > "What is the meaning of 'rigorous probabilistic interpretation' in Line 128? Does that mean the new P_B(df) is a properly defined functional posterior?"
>
> We mean that $\frac{1}{\lambda} d_B(f, F)$ can be interpreted as a negative log likelihood encoding that we observe $d_B(f, F) = 0$ with Laplace distributed noise (with parameter $\lambda$). The proposition shows that the optimization problem is the corresponding MAP estimation problem.
>
> > "The objective function is similar to the kernel ridge regression. Can you please explain the difference?"
>
> Good observation! The kernel ridge regression (KRR) can be seen as the maximum a-posteriori of a GP.
> Thus, the first part of our algorithm, which learns an RKHS-regularized neural network, bears some similarity with KRR.
> However, KRR is a nonparametric method wherease the first part of our algorithm optimizes the parameters of a neural network (this is motivated by the generalization performance of deep neural networks, as mentioned in the abstract).
> Further, the second part of our algorithm estimates uncertainties of the neural network parameters, which has no analog in KRR.

---

> > ### Comment · Reviewer_ospn · 2024-08-12
> >
> > Thanks for the authors' response!

---

### Official Review · Reviewer_wqcS · 2024-07-15

**Soundness:** 3
**Presentation:** 4
**Contribution:** 4
**Rating:** 8
**Confidence:** 4

**Summary:**

The paper proposes a new method (FSP-Laplace) to place priors in the function space of deep neural nets and develop scalable Laplace approximations on top of it. The ideas are inspired in MAP estimation theory and the fact that an objective function that is actually regularising the neural network in the function space can be developed. Additionally, to face scalability and mitigate the issues of the computational costs in calculating large curvature matrices, the authors introduce methods from matrix-free linear algebra (i.e. Lanczos). Experimental results show that the FSP-Laplace method works well in such problems were makes sense to encode information in the form of a GP prior, and in general, overcomes the standard Laplace approximation.

**Strengths:**

The paper is well-written and in general, well polished to give the right and clear details of a methodology that could be very difficult to follow otherwise. In that regard, the effort on synthesis and clarification is a big strength. The quality of the manuscript is therefore high (including the technical methods considered and the problem faced).

Getting more in detail, I particularly like the spirit of the work and the idea of building a Laplace approximation directly on the functional space of the deep NN. Despite I don't see exactly how feasible could be to compute the potential \phi and Eq. (3.1), I see what the authors actually do with the idea of the constraint in Propositions 1 and 2.

Despite I have a little question here, I also see a point of strength in the way that the local linearisation fixes the problem of having valid unnormalized negative log-densities.

The use of the matrix-free methods is also thorough and despite the fact that it adds an extra point of complexity, it is clear what the advantages are and how it positively affects the scalability and performance of the method in practice. Last but not least, experimental results seem to show a good performance, and despite being somehow short, they are kind of sufficient and standard for the Laplace approximation considered and also in comparison with the rest of Laplace SOTA papers in the last years.

**Weaknesses:**

Even if I (honestly) don't find many weaknesses in the work, I think that the following points make the approach a bit weaker:

- Context points in section 3.2 are kind of an obscure part of the Algorithm to me. Despite it is indicated that the best strategy is to use iid samples and that it is an effective way to compute the FSP-Laplace objective, I think a bit of related work and connections in this point could be needed. In general, it reminds me of inducing points in GPs and kind of pseudo-coresets. I also see that some limitations or issues with this sampling and the number of context points could appear, and somehow it is not really considered in the manuscript (in my opinion).

- For instance, Laplace Redux [Daxberger 2021] states clearly the advantage of using Laplace approximation that it is feasible for both regression and classification problems, once the MAP estimate is obtained. In the case of FSP-Laplace, it is not clear to me if the work is also fully applicable to both classification and regression problems in an easy way, to which likelihood models, and if doing that makes things more difficult on the FSP-Laplace evaluation part or the Laplace approximation side. Perhaps, a bit more of clarification in this direction could be useful.


**References.**

[Daxberger 2021] -- Laplace Redux – Effortless Bayesian Deep Learning, NeurIPS 2021.

**Questions:**

**Q1** -- In L201, it is mentioned that there is an observation on the fact that the trace of the posterior is 'always' upper-bounded by the trace of the prior covariance. Due to this, a little condition is imposed as a remedy. Is this an empirical observation? it is always true? or is there a proof for this?

**After rebuttal comments:** Thanks to the authors for their detailed comments and clarifications to the few questions and points I raised in my first review. I am, in general, excited with the contributions that the paper brings to the community, particularly on Bayesian NNs and topics related to the (linearised) Laplace approximation. I do think that the paper is strong and should be accepted, so I thereby increase my score to 8.

**Limitations:**

Yes

---

> ### Author Rebuttal · Authors · 2024-08-07
>
> We thank the reviewer for their time and constructive feedback. We are glad that they found that the "quality of the manuscript is [...] high", that they "like the spirit of the work" and that our method has "good performance".
>
> > "Context points in section 3.2 are kind of an obscure part of the Algorithm to me. [...] I think a bit of related work and connections in this point could be needed. In general, it reminds me of inducing points in GPs and kind of pseudo-coresets. I also see that some limitations or issues with this sampling and the number of context points could appear, and somehow it is not really considered in the manuscript (in my opinion)."
>
> Methods for regularizing neural networks in function space always rely on context points to evaluate the regularizer (Sun et al. 2019, Tran et al. 2022, Rudner et al. 2023). Popular strategies include uniform sampling from a distribution over a bounded subset of the feature space (Tran et al. 2022, Rudner et al. 2023), from the data (Sun et al., 2019), and from additional (possibly unlabeled) datasets (Rudner et al. 2023).
>
> The goal of the context points is to regularize the neural network on a finite set of input locations which includes the training data and any point where we would like to evaluate the model. During the MAP estimation phase of our method (see Algorithm 2), context points are different from inducing points in GPs: context points are resampled at each update, unlike inducing points of a GP, which are optimized or kept fixed. When computing the posterior covariance, however, context points indeed share some similarity with inducing points in a GP: at this point, they are fixed and define where we regularize the model.
> Unlike pseudo-coresets, context points do not aim to summarize the training data but rather to cover a finite set of input locations that includes the training data and any point where we wish to evaluate the model.
>
> Relying on a set of context points is fine for datasets with low-dimensional features, which are precisely the cases for which we have prior beliefs in the form of a GP prior (GP priors are much harder to specify with high-dimensional data). Our FSP-Laplace method was precisely designed with these scenarios in mind and the scalable matrix-free linear algebra methods can perfectly cope with a very large number of context points (> 20'000). For high-dimensional data, using context points is manageable if we have prior knowledge about where the training data lies (for example, a manifold in the ambient feature space). We can then use a set of context points that has approximately the same support, as shown in our image classification examples where we use a set of context point from the KMNIST dataset. Nevertheless, Table D.2 in the PDF linked in the general rebuttal also shows that our method remains competitive on MNIST image classification when using context points drawn from a uniform distribution during training (RND). Due to limited time during the rebuttal period, results for Fashion MNIST are not yet available (N.As in Table D.2) and we will update the reviewers during the discussion phase with the results. Another example where prior beliefs can often be formulated in the form of a GP is scientific data (see, e.g., our ocean current experiment). For many scientific data, context points could also be generated by a simulation that does not need to be very accurate (since it is only meant to generate points in the vicinity of the true data distribution).
>
> A sufficient number of context points is necessary to capture the beliefs specified by the GP prior (see Figure D.3 and D.4 in the PDF link to the general rebuttal). If the number of context points is too small, the beliefs are not accurately captured, and the function draws will not look like the prior but the uncertainty should not collapse (see Figures D.3 and D.4 in PDF in the general rebuttal).
>
> > In the case of FSP-Laplace, it is not clear to me if the work is also fully applicable to both classification and regression problems in an easy way, to which likelihood models, and if doing that makes things more difficult on the FSP-Laplace evaluation part or the Laplace approximation side.
>
> FSP-Laplace works in exactly the same settings as the standard Laplace approximation, e.g., for classification and regression settings using Gaussian and Categorical likelihoods, respectively (see regression and classification examples in Section 4.1). Just like Laplace, we require that the likelihood function is twice differentiable (we need to compute its Hessian). Note that for the classification setting with $c$ classes, we place a GP prior on every logit such that we have $c$ priors. Therefore, $M$ is the $p \times rc$ matrix containing the concatenation of the $c$ $(J_i^T L_i)_{i=1}^c$ matrices in line 4 of Algorithm 1. We will make this clearer in the camera-ready version.
>
> > "In L201, it is mentioned that there is an observation on the fact that the trace of the posterior is 'always' upper-bounded by the trace of the prior covariance. Due to this, a little condition is imposed as a remedy. Is this an empirical observation? it is always true? or is there a proof for this?"
>
> This follows from the expression of the covariance of the posterior in a Gaussian process, which is the prior covariance minus a term corresponding to the conditioning on the training data. The latter is always non-negative.
>
> > "I don't see exactly how feasible could be to compute the potential \phi and Eq. (3.1)"
>
> Intuitively, the potential can be understood as the negative log-likelihood.

---

### Official Review · Reviewer_Lja8 · 2024-07-17

**Soundness:** 3
**Presentation:** 3
**Contribution:** 2
**Rating:** 5
**Confidence:** 4

**Summary:**

This paper proposes a method to calculate the Laplace approximation of the posterior of neural networks with a prior defined directly in the functional space (Gaussian processes). Due to the absence of Lebesgue densities in infinite-dimensional functional spaces, the notion of weak mode is used to obtain an analogy to MAP estimation in the functional space. Through model linearization, the Laplace approximation is performed at the MAP parameter of the neural network. Experiments on both synthetic and real-world data demonstrate that the proposed method effectively captures the epistemic uncertainty in function fitting and classification.

**Strengths:**

- The paper addresses the posterior approximation in Bayesian deep learning, an important area for quantifying predictive uncertainty of models. The proposed method is well-motivated by the limitations of placing prior distributions over the parameter space of neural networks. The techniques used in this paper are solid.
- The paper is generally very well-written, with clear and easy-to-follow notation.

**Weaknesses:**

My main concern is the relation between this paper and Sun et al. (2019) [8]. Sun et al. also consider Bayesian inference of function posteriors with a functional prior directly defined in the function spaces (e.g., Gaussian processes). The main difference seems to be that Sun et al. perform fully Bayesian inference, while this work aims to obtain a MAP estimate followed by a Laplace approximation to obtain the posterior. Sun et al.'s work should be included as a baseline, and more discussion on the relation between the two works is needed.

Similar to the difficulty in evaluating the functional KL divergence in Sun et al., when evaluating the "prior probability" (i.e., RKHS norm) of $\\mathbf\{f\}$, a set of context points must be specified to approximate it. My second concern is that this sampling strategy could be inefficient and impractical for high-dimensional datasets. For 1D or 2D datasets, the context points can be sampled uniformly or fixed to grid points. However, for higher-dimensional cases, this approach becomes less effective. This is a critical issue for the proposed method, as in high-dimensional spaces, there is little chance that the context points will adequately cover the support of the data distributions. For example, using KMNIST as the context points for MNIST may not always be feasible in general cases.

Minor:
- L 89, Section 3.1 -> Section 3.2.
- L271, L533, L597: References are missing.

**Questions:**

- What will happen if the same context points are used for both training and approximating the Hessian?

Additionally, please also see my questions raised in the Weakness section regarding the relation between this work and Sun et al., 2019, and the challenges of the context point sampling strategy for high-dimensional datasets.

**Limitations:**

Yes.

---

> ### Author Rebuttal · Authors · 2024-08-07
>
> We thank the reviewer for their useful feedback and time. We are glad that they found our "method is well-motivated", our "techniques [...] are solid" and that the paper was "very well-written" and "easy-to-follow".
>
> > "My main concern is the relation between this paper and Sun et al. (2019) [8]. Sun et al. also consider Bayesian inference of function posteriors with a functional prior directly defined in the function spaces (e.g., Gaussian processes). The main difference seems to be that Sun et al. perform fully Bayesian inference, while this work aims to obtain a MAP estimate followed by a Laplace approximation to obtain the posterior. Sun et al.'s work should be included as a baseline, and more discussion on the relation between the two works is needed."
>
> While Sun et al. (2019) and our method both specify GP priors on the function represented by the neural network, Sun et al. use variational inference (VI), whereas our method uses the Laplace approximation. VI approximates the posterior by a parametric distribution that maximizes the ELBO (Eq. 3 in Sun et al.), while the Laplace approximation approximates the posterior by a Gaussian centered at the MAP estimate of the parameters. Note that the Laplace approximation and VI are both parametric *approximate* posterior inference methods that are commonly used in the machine learning and statistics literature. Neither one of them can be considered more "fully Bayesian" than the other. We agree that a comparison with Sun et al. is interesting, and we therefore include experiments using this method (labeled "FVI") in our overall rebuttal above. Our method FSP-Laplace generally outperforms FVI. This can possibly be explained by a complication in the FVI method: evaluating the ELBO requires access to the density of the variational distribution, which is not available in function space. FVI addresses this issue with implicit score function estimators, which introduce another approximation and makes FVI difficult to use in practice (Ma and Hernández-Lobato, 2021). On a more theoretical level, note that the KL divergence between measures (Eq. 4 in Sun et al.) is actually infinite when using GP priors (Burt et al., 2020) and the function-space ELBO (Eq. 3 in Sun et al.) is undefined. Our Laplace approximation does not suffer from either of these issues and approximates a well-defined objective.
>
> > "Similar to the difficulty in evaluating the functional KL divergence in Sun et al., when evaluating the 'prior probability' (i.e., RKHS norm) of $f$, a set of context points must be specified to approximate it. My second concern is that this sampling strategy could be inefficient and impractical for high-dimensional datasets. For 1D or 2D datasets, the context points can be sampled uniformly or fixed to grid points. However, for higher-dimensional cases, this approach becomes less effective. This is a critical issue for the proposed method, as in high-dimensional spaces, there is little chance that the context points will adequately cover the support of the data distributions. For example, using KMNIST as the context points for MNIST may not always be feasible in general cases."
>
> Methods that regularize a neural network in function space (both for approximate Bayesian inference or regularized empirical risk minimization) always use a set of context points to evaluate the regularizer (Sun et al. 2019, Tran et al. 2022, Rudner et al. 2023). As you say, relying on a set of context points is fine for datasets with low-dimensional features, which are precisely the cases for which we have prior beliefs in the form of a GP prior (GP priors are much harder to specify with high-dimensional data). Our FSP-Laplace method was precisely designed with these scenarios in mind and the scalable matrix-free linear algebra methods can perfectly cope with a very large number of context points (> 20'000). For high-dimensional data, using context points is manageable if we have prior knowledge about where the training data lies (for example, a manifold in the ambient feature space). We can then use a set of context points that has approximately the same support, as shown in our image classification examples where we use a set of context points from the KMNIST dataset. Nevertheless, Table D.2 in the PDF linked in the general rebuttal also shows that our method remains competitive on MNIST image classification when using context points drawn from a uniform distribution during training. Due to limited time during the rebuttal period, results for Fashion MNIST are not yet available (N.As in Table D.2) and we will update the reviewers during the discussion phase with the results. Another example where prior beliefs can often be formulated in the form of a GP is scientific data (see, e.g., our ocean current experiment). For many scientific data, context points could also be generated by a simulation that does not need to be very accurate (since it is only meant to generate points in the vicinity of the true data distribution). We will add more details in the camera-ready paper.
>
> > "What will happen if the same context points are used for both training and approximating the Hessian?"
>
> This is fine. It is only important that the set of context points covers a finite set of input locations containing all the training data as well as any points where we would like to evaluate the model. During training, we can amortize the coverage by resampling the context points at each update step (step 5 in Algorithm 2). When computing the covariance, we can no longer amortize and must use a set of points that covers the space well, such as a low discrepancy sequence (Latin hypercube sampling, for example).

---

### Author Rebuttal · Authors · 2024-08-07

We thank all reviewers for their time and feedback. Some reviewers requested a comparison with FVI (Sun et al., 2019) and more details on the effect of the context points. We ran these additional experiments and wish to share them with all the reviewers.

**Comparison with FVI (Sun et al., 2019):** We ran our suite of experiments on FVI and report results in the Tables D.1 to D.3 and Figures D.1 and D.2 in the linked PDF file.
Due to limited time during the rebuttal period, we are unable to provide results for FSP-Laplace using the RND context points on Fashion MNIST (N.As in Table D.2). We will update the reviewers with the results during the discussion phase.

We find that FVI is less accurate than FSP-Laplace and baselines on the Mauna Loa $CO_2$ forecasting task (see Table D.1). Due to limited space in the 1 page PDF, we do not show Figure C.10 updated with FVI. On the ocean current modeling task, FVI performs similarly to our FSP-Laplace method in terms of mean squared error but strongly underestimates the predictive variance (see Figure D.1) and therefore incurs a lower test expected log-likelihood than FSP-Laplace (see Table D.1). We also find that FSP-Laplace shows higher expected log-likelihood and accuracy on the image classification tasks and produces models with lower expected calibration error (ECE) than FVI (see Table D.2). FSP-Laplace also obtains higher OOD detection accuracy than FVI when using the KMNIST context point distribution. Preliminary results using context points uniformly sampled from a subset of the features space (RND) on MNIST, also show improved performance compared to FVI. On the Bayesian optimization task, we find that FSP-Laplace converges faster than FVI on Branin, PDE, and BNN, that FVI converges faster than FSP-Laplace on Polynomial, and that both methods perform similarly on Ackley and Hartmann (see Figure D.2).
Finally, on the UCI regression tasks in Table D.3, we find that FSP-Laplace and FVI perform similarly in terms of expected log-likelihood with a slight advantage for FSP-Laplace (mean rank of 1.545 vs. 1.700). However, FSP-Laplace almost systematically performs best in terms of out-of-distribution detection (mean rank of 2.182 vs. 3.000).

**Effect of context points:** We show additional results demonstrating the behavior of our model in the low context point regime.

Figure D.3 shows the effect of the number of context points on a 1-dimensional regression task with GP priors equiped with a RBF and a Matern 1/2 kernel. Figure D.4 shows the same experiment but on a 2-dimensional classification task. The M context points are randomly sampled uniformly during training, and we use the Halton low discrepancy sequence as context points to compute the covariance.

We show that even with a very small number of context points (M=3 and M=5), our model still produces useful uncertainty estimates even if it cannot accurately capture the beliefs specified by the prior. We also note that our method requires more context points to capture the beliefs of rougher priors than smooth priors (see Figure D.3).

Table D.2 shows the results of FSP-Laplace when using the Kuzushiji-MNIST dataset (KMNIST) and draws from a uniform distribution (RND) as context points during training. We use the Halton low-discrepancy sequence as context points to compute the covariance. Due to limited time during the rebuttal period, we are unable to provide results for FSP-Laplace using the RND context points on Fashion MNIST (N.As in Table D.2). We will update the reviewers with the results during the discussion phase. On MNIST, we find that FSP-Laplace with RND context points obtains similar expected log-likelihood, accuracy, and expected calibration error as with the KMNIST context point distribution. OOD detection accuracy is lower than with context points from KMNIST but remains very competitive with respect to baselines.

---

### Decision · Program_Chairs · 2024-09-25

**Decision:**

Accept (poster)

**Comment:**

The submission considers using a function space prior in place of standard weight priors for the Laplace approximation in neural networks. After several approximations, this boils down to a tractable regularisation that enforces the neural network's prior predictive distribution to match that of a Gaussian process prior at a small number of context points. The empirical results are generally promising, showing the utilities of the new method. The suite of experiments is, however, not as comprehensive as recent Bayesian deep learning / Laplace papers.

A reviewer raised a substantial concern, resulting in an unsettled debate between the reviewer and authors in the rebuttal. The issue is the theoretical justification of Laplace and its independence from the prior. The AC noted the concern is nitpickingly technically correct. However, what the authors proposed and the approximations to make things tractable, similar to other recent `Laplace` methods, is approximate and no longer strictly Laplace and seems to give good empirical results. Other reviews are generally positive about the submission. The AC thus suggests acceptance.

One thing that the reviewers have surprisingly missed is Rudner et al's tractable functional VI scheme (NeurIPS 2022). The AC encourages the authors to compare FSP-Laplace to this.